# Preclinical characterization and target validation of the antimalarial pantothenamide MMV693183

Laura E. de Vries [1,25], Patrick A. M. Jansen[2], Catalina Barcelo[3], Justin Munro[4], Julie M. J. Verhoef [1], Charisse Flerida A. Pasaje [5], Kelly Rubiano[6], Josefine Striepen [6], Nada Abla[3], Luuk Berning [7], Judith M. Bolscher[7], Claudia Demarta-Gatsi [3], Rob W. M. Henderson[7], Tonnie Huijs[7], Karin M. J. Koolen[7], Patrick K. Tumwebaze[8], Tomas Yeo [6], Anna C. C. Aguiar[9], Iñigo Angulo-Barturen [10], Alisje Churchyard [11], Jake Baum [11], Benigno Crespo Fernández[12], Aline Fuchs[3], Francisco-Javier Gamo[12], Rafael V. C. Guido [9], María Belén Jiménez-Diaz[10], Dhelio B. Pereira [13], Rosemary Rochford[14], Camille Roesch [15,16], Laura M. Sanz [12], Graham Trevitt[17], Benoit Witkowski[15,16], Sergio Wittlin[18,19], Roland A. Cooper[20], Philip J. Rosenthal[21], Robert W. Sauerwein[1,7], Joost Schalkwijk [2], Pedro H. H. Hermkens[22], Roger V. Bonnert[3], Brice Campo [3], David A. Fidock [6,23], Manuel Llinás [4,24], Jacquin C. Niles [5], Taco W. A. Kooij [1,26 ✉] & Koen J. Dechering [7,26 ✉]

Drug resistance and a dire lack of transmission-blocking antimalarials hamper malaria elimination. Here, we present the pantothenamide MMV693183 as a first-in-class acetyl-CoA synthetase (AcAS) inhibitor to enter preclinical development. Our studies demonstrate attractive drug-like properties and in vivo efficacy in a humanized mouse model of *Plasmodium falciparum* infection. The compound shows single digit nanomolar in vitro activity against *P. falciparum* and *P. vivax* clinical isolates, and potently blocks *P. falciparum* transmission to *Anopheles* mosquitoes. Genetic and biochemical studies identify AcAS as the target of the MMV693183-derived antimetabolite, CoA-MMV693183. Pharmacokinetic-pharmacodynamic modelling predict that a single 30 mg oral dose is sufficient to cure a malaria infection in humans. Toxicology studies in rats indicate a > 30-fold safety margin in relation to the predicted human efficacious exposure. In conclusion, MMV693183 represents a promising candidate for further (pre)clinical development with a novel mode of action for treatment of malaria and blocking transmission.

A full list of author affiliations appears at the end of the paper.

Malaria remains a significant global infectious disease, caused by parasites of the genus *Plasmodium*. In the past two decades there was a major decline in malaria cases and deaths, however, this progress has stalled, emphasizing the need for new interventions[1]. Drug resistance against many front-line therapies is emerging and spreading around the world, threatening the efficacy of these drugs[1–3]. There is an urgent need for new therapeutics to combat the spread of resistance and to progress towards malaria elimination. Target product profiles and target candidate profiles were developed to guide the discovery and clinical development of new antimalarials[4]. Current approaches for new malaria treatments aim for a combination of two or more inexpensive, potent, fast-acting molecules that act on multiple parasite stages and provide a single dose cure[4]. Compounds with new modes of action are favored since no pre-existing resistance in the field would be expected.

Coenzyme A (CoA) is required for numerous processes within the cell, including lipid synthesis, protein acetylation, and energy supply, and it is highly conserved among prokaryotes and eukaryotes[5]. *Plasmodium* parasites rely on this pathway by uptake of the essential nutrient pantothenic acid (pantothenate or vitamin B5) that is converted into CoA in five enzymatic reactions[6–8]. The CoA biosynthesis pathway in *Plasmodium* species has been considered a potential drug target since the discovery of the antimicrobial activity of pantothenic acid derivatives in the 1940s[9]. Different libraries of pantothenic acid derivatives have been synthesized since[9], however, due to poor stability in human serum they have never been developed into clinical candidates[10–12].

In the past decade, the focus has been on developing stable pantothenamides (PanAms), in which the terminal carboxyl group of pantothenic acid is replaced by amides[12–14], including our recently synthesized PanAm with an inverted-amide bond (iPanAms)[15]. These stabilized iPanAms have micromolar activity against gram-positive bacteria[16,17], and an IC$_{50}$ of 0.95 μM against *Toxoplasma gondii*[18]. Furthermore, PanAms are highly potent against pathology-causing asexual blood stages and transmittable gametocytes of *Plasmodium falciparum*, consistent with the essentiality of several enzymes of the CoA pathway in both life-cycle stages[19–21]. This indicates their potential to be developed into antimalarials that target a wholly novel pathway thereby curing the disease and blocking transmission to the mosquito host.

The exact mechanism of action of PanAms has been debated extensively, with pantothenate uptake, pantothenate kinase or CoA-utilizing processes as possible targets[7,8,15,21–23]. The latest studies have indicated that the latter is the likely target. PanAms are metabolized by three enzymes of the CoA biosynthesis pathway, and form analogs of CoA pathway metabolites, including 4'P-PanAm, dP-CoA-PanAm and CoA-PanAm[15,21,22]. A combination of biochemical and genetic approaches have demonstrated that these antimetabolites likely target the downstream enzymes acetyl-CoA synthetase (AcAS; PF3D7_0627800) and acyl-CoA synthetase 11 (ACS11; PF3D7_1238800), thereby inhibiting the synthesis of acetyl-CoA[15]. However, definitive proof of drug-enzyme interactions remained elusive.

Here, we describe the generation of the novel pantothenamide MMV693183 and demonstrate that its CoA-PanAm metabolite targets AcAS. Moreover, MMV693183 has improved potency and metabolic stability, and a prolonged killing effect in a humanized mouse model of *P. falciparum* in comparison to previously described PanAms, and thus meets the requirements for further (pre)clinical development[4].

## Results

**MMV693183 is a potent antimalarial drug candidate.** We recently synthesized a novel class of iPanAms with an inverted-amide bond that resulted in compound MMV689258 (**1**) with a limited predicted half-life in humans[15]. Therefore, we continued chemical optimization, and a subseries of potent compounds with a (hetero)aromatic ring directly coupled to the inverted amide was generated[24]. In general, nitrogen-containing heteroaromatics such as pyrrole and pyridine showed relatively poor potency (Supplementary Tables 1 and 2). Indoles exerted reasonable activity, with attachment at the 3-position slightly dominant over the 2-position. The most active heterocycle was a 3-substituted benzofuran with an IC$_{50}$ of 3.9 nM against *P. falciparum* asexual blood-stage parasites. Exploration of phenyl substitutions showed a preference for F, Cl, and CN over polar groups like amines and sulphones. Combinations showed best activity for 2,3 and 1,4 substitutions or 1,3,4-trifluoro or 1,3-fluoro, 4-chlorine. A subset of these iPanAms with an average IC$_{50}$ of < 16 nM against asexual blood stages was further profiled and five of these showed lower clearance in human primary hepatocytes compared to MMV689258, potentially leading to a longer half-life in humans. In addition, they showed activity against sexual blood-stage *P. falciparum* parasites with an average IC$_{50}$ value of ≤ 31 nM except for one PanAm that was not active up to 1 μM against gametocytes (Table 1).

As previous experiments have shown that the in vivo efficacy of MMV689258 outlives its plasma exposure[25], we performed a single dose survey to investigate to what extent the new iPanAms show a similar effect. To this end, humanized mice were infected with *P. falciparum* and treated with a single dose of 50 mg/kg of PanAm by oral gavage. For all compounds, blood concentrations decreased rapidly over time and were either below or near the detection limit (5 ng/ml) after 24 h. In spite of the rapid elimination, MMV693183 (**2**), MMV884962 (**3**), and MMV1542001 (**4**) reduced parasitemias below detectable levels over the course of three days, while parasitemias were not fully cleared upon treatment with MMV689258, MMV693182 (**5**), and MMV976394 (**6**) (Fig. 1a; Table 1; Supplementary Table 3). Attempts to obtain crystalline forms of the six iPanAms, important for future drug formulation in tablets, were only successful for MMV689258, MMV693183, and MMV693182 (Supplementary Fig. 1), and resulted in favorable melting temperatures for the latter two compared to MMV689258 (Table 1). MMV693183 was selected as an advanced lead compound for further study, as it combined all improved characteristics. Furthermore, it was highly soluble in PBS, as well as in fasted- and fed-state simulated intestinal fluids (7.1, 9.2, 9.1 mg/ml, respectively) (Supplementary Table 4). MMV693183 was also chemically stable after storage under stress conditions (40 °C, 75% relative humidity in an open and closed container or at 60 °C) (Supplementary Fig. 2).

Asexual blood-stage parasites treated with MMV693183 in a parasite reduction rate (PRR) assay showed rapid killing activity. Parasitemia was reduced within 24 h and to below the detection limit within 48 h (Supplementary Fig. 3). This profile is similar to artemisinins[26] that constitute the fastest-acting class of clinical antimalarials available to date. MMV693183 was active against both early and late asexual blood stages (Supplementary Fig. 4), but was not efficacious against liver stages (Supplementary Fig. 5), similar to previous findings[15]. Treatment of gametocytes 24 h before feeding to *A. stephensi*, inhibited oocyst formation with an IC$_{50}$ value of 38 nM (Fig. 1b), but treatment with 1 μM directly at the time of the mosquito feeds did not inhibit midgut infection (Fig. 1c), confirming the gametocytocidal mode of action. MMV693183 specifically inhibited female gametocyte activation with an IC$_{50}$ value of 12 nM, whereas male gamete formation was inhibited much less with an IC$_{50}$ value of 1 μM (Fig. 1d). We also performed ex vivo activity assays against *P. falciparum* and *P. vivax* field isolates from Uganda and Brazil and against artemisinin-resistant *P. falciparum* Cambodian field isolates

**Table 1 Physicochemical characteristics and in vitro activities of pantothenamides.**

| Compound | Chemical structure | Asexual blood stage IC$_{50}$ range (N) | Gametocyte IC$_{50}$ range (N) | Human hepatocyte CL$_{int}$ (µl/min/ 10$^6$ cells) | Molecular Weight (g/mol) | Crystalline | Melting temperature (°C) |
|---|---|---|---|---|---|---|---|
| (1) MMV689258 | | 5* nM | 12* nM | 0.8 | 354.4164 | Yes | 48.1 |
| (2) MMV693183 | | 2.1-2.8 (4) nM | 17.8-38.8 (3) nM | 0.4 | 362.3441 | Yes | 91.6 |
| (3) MMV884962 | | 7.4-12 (2) nM | 2.2-5.9 (2) nM | 0.3 | 333.3822 | No | N/A |
| (4) MMV1542001 | | 1.9-2.4 (2) nM | 2.0-9.9 (2) nM | 0.5 | 326.3632 | No | N/A |
| (5) MMV693182 | | 6.2-7.5 (2) nM | 21.4-32.9 (2) nM | 0.2 | 351.3727 | Yes | 107.9 (1-1) 112.7 (1-2) |
| (6) MMV976394 | | 6.2-24.9 (2) nM | > 1 µM (2) | 0.2 | 337.3461 | No | N/A |

IC$_{50}$ values were determined using a nonlinear regression with four-parameter model and the least-squares method to find the best fit. The IC$_{50}$ range is shown from two to four independent experiments (N) with technical duplicates measured in a 72-h asexual growth assay or sexual viability assay. Crystallization of MMV693182 resulted in two polymorphs. *Data retrieved from Schalkwijk et al.[15]. N/A: not applicable. Source data are provided as Source Data file.

adapted to in vitro culture. Encouragingly, all isolates were sensitive to the drug, with low nanomolar IC$_{50}$ values (Fig. 1e; Supplementary Fig. 6).

**A role for AcAS in the mechanism of action of MMV693183.** In vitro evolution and whole-genome analysis (IVIEWGA) experiments identified a single point mutation in *AcAS* (PF3D7_0627800) resulting in a T648M amino acid change (dT648M) in *P. falciparum* Dd2-B2 and NF54 strains pressured at sublethal concentrations of MMV693183 (Supplementary Tables 5 and 6). Exposing different inocula of Dd2-B2 parasites to the compound cleared parasites within a few days up to an inoculum size of 1×10$^8$ and yielded no parasite recrudescence when followed for 60 days. At an inoculum of 1×10$^9$ Dd2-B2 parasites did not fully clear and MMV693183 resistant parasites emerged over the course of 30 days (Supplementary Table 5,

Supplementary Fig. 7). These resistant parasites showed a 13-77× IC$_{50}$ shift against MMV693183 in an asexual blood-stage growth assay and also resulted in resistant gametocytes (> 50× IC$_{50}$ shift) (Fig. 2a) that were able to transmit to mosquitoes (Supplementary Fig. 8a-c). CRISPR-Cas9 engineering of this mutation in wild-type parasites (cT648M) confirmed the resistance phenotype observed for the T648M mutation (dT648M) in both asexual and sexual blood stages (49× and > 50× IC$_{50}$ shift, respectively), and conferred cross-resistance to previously generated iPanAms (Fig. 2a, Supplementary Fig. 9)[15]. In addition, mutant parasites with a previously described T627A mutation in AcAS[15] were also resistant to MMV693183 (Fig. 2a). Metabolomic profiling showed that MMV693183 was metabolized into three CoA-precursors (Fig. 2b, bottom panel, Supplementary Fig. 10) and reduced acetyl-CoA and 4-phosphopantothenate levels in infected RBCs in a dose-dependent manner (Fig. 2b, upper panel). This is in line

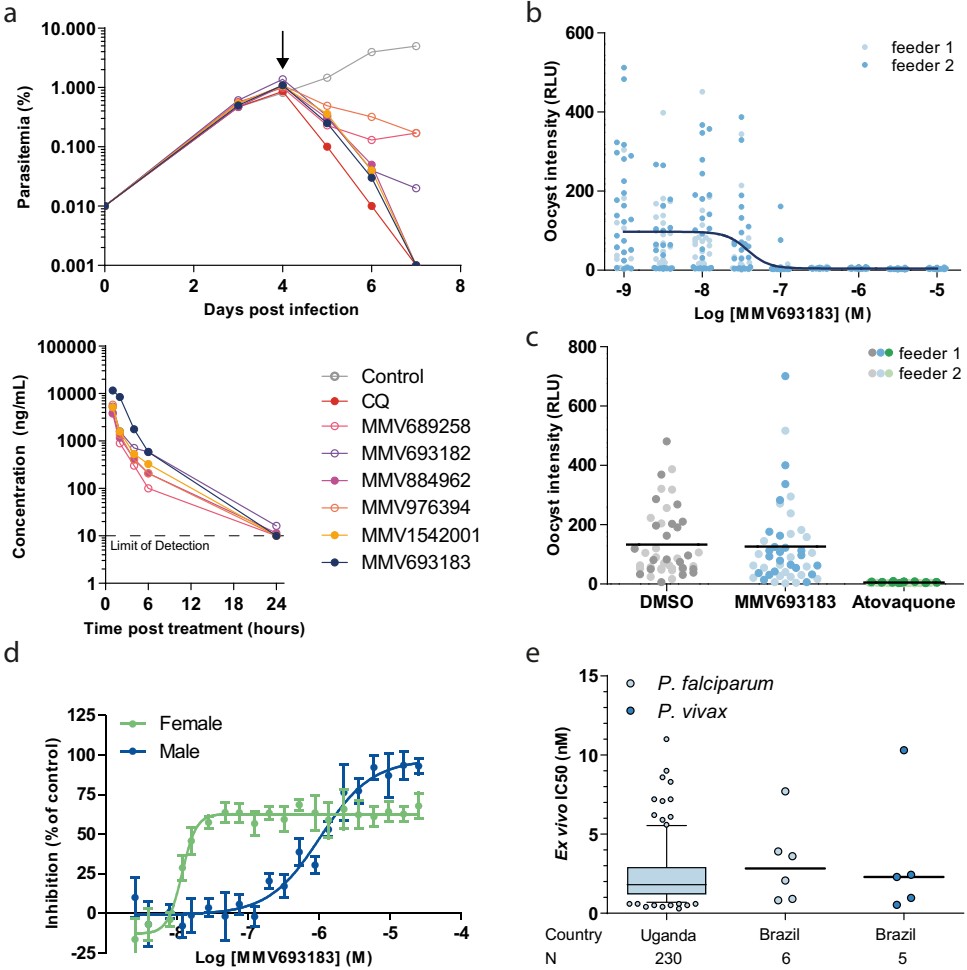

**Fig. 1 Antimalarial activity of the pantothenamide MMV693183. a** In vivo activity of novel pantothenamides. NSG mice were infected with *P. falciparum* on day 0. Mice were treated with pantothenamides by oral gavage (50 mg/kg) ($N = 2$/compound) on day 4 (arrow) and parasitemia was quantified every day from day 3 onwards (top panel). The corrected concentration of pantothenamides in blood is indicated in the bottom panel. **b** The activity of MMV693183 on *P. falciparum* (NF54-HGL) stage V gametocytes treated for 24 h before mosquito feeding in a single experiment with two replicates (feeder 1, 2). Oocyst intensity was measured by luminescence eight days after the feed. **c** Oocyst intensities in mosquito midguts when *P. falciparum* (NF54-HGL) stage V gametocytes were exposed to 1 µM MMV693183, 100 nM atovaquone, or 0.1% DMSO within the mosquito blood meal. Oocyst intensity was quantified by luminescence eight days after feeding in a single experiment with two replicates (feeder 1, 2). **d** Dual gamete formation assay upon treatment of female or male gametocytes with MMV693183 in four independent experiments (±SD). Typically, 150-250 exflagellation centers or 2000-3000 female gametes per field were recorded in the negative controls. **e** Ex vivo activity of MMV693183 against field isolates of *P. falciparum* from Uganda ($N = 230$) in a parasite growth assay and against field isolates of *P. falciparum* ($N = 6$) and *P. vivax* ($N = 5$) from Brazil in a schizont maturation assay. Median $IC_{50}$ values and 5-95 percentile are shown in a Box-Whisker plot for the field isolates from Uganda. Minimum: 0.3 nM, maximum: 11.0 nM, box: 25-75 percentile, whiskers: 5-95 percentile. CQ, chloroquine. Source data are provided as Source Data file.

with previous observations of PanAm antimetabolite generation and suggests inhibition of AcAS function.

To further confirm a role for AcAS in the mechanism of action to MMV693183, we used the previously generated AcAS conditional knockdown parasite line[27] based on the TetR-DOZI system that allows repression of translation of the target gene when the drug anhydrotetracycline (aTc) is removed[28,29]. We cultured conditional knockdown AcAS parasites (AcAS-cKD) and control knockdown parasites (control-cKD) in low aTc (1.5 or 0 nM aTc, respectively) or high aTc (500 nM) conditions and exposed them to different doses of MMV693183. The $IC_{50}$ for the AcAS-cKD parasite line decreased 5-fold upon knockdown conditions (low aTc), showing hypersensitivity of these parasites to the compound, while there was no difference in sensitivity for the negative knockdown control (Fig. 2c). This supports our hypothesis that PanAms inhibits *P. falciparum* growth in an AcAS-dependent manner.

**CoA-PanAm targets AcAS.** To provide conclusive evidence that iPanAms directly bind to AcAS, thereby inhibiting AcAS activity, we used the recently developed cellular thermal shift assay (CETSA)[30,31] to test the thermal stability of AcAS upon treatment with iPanAms. Rabbits were immunized with a recombinant AcAS fragment and the induced anti-serum was used to detect AcAS (Supplementary Fig. 11). Following incubation of asexual blood-stage lysate with compounds for 30 min, neither the parent compound MMV693183 nor its derivative 4'P-MMV693183 affected AcAS stability. However, the CoA-MMV693183 metabolite clearly stabilized AcAS upon temperature increase (Fig. 3a). This supports the notion that iPanAms form active CoA-PanAm antimetabolites that target AcAS. To provide further evidence that CoA-PanAm targets AcAS, we immunopurified AcAS from wild-type and cT648M parasite lysates and established an AcAS activity assay. CoA-MMV693183 inhibited wild-type AcAS activity with an $IC_{50}$ of 300 nM, while it only weakly inhibited

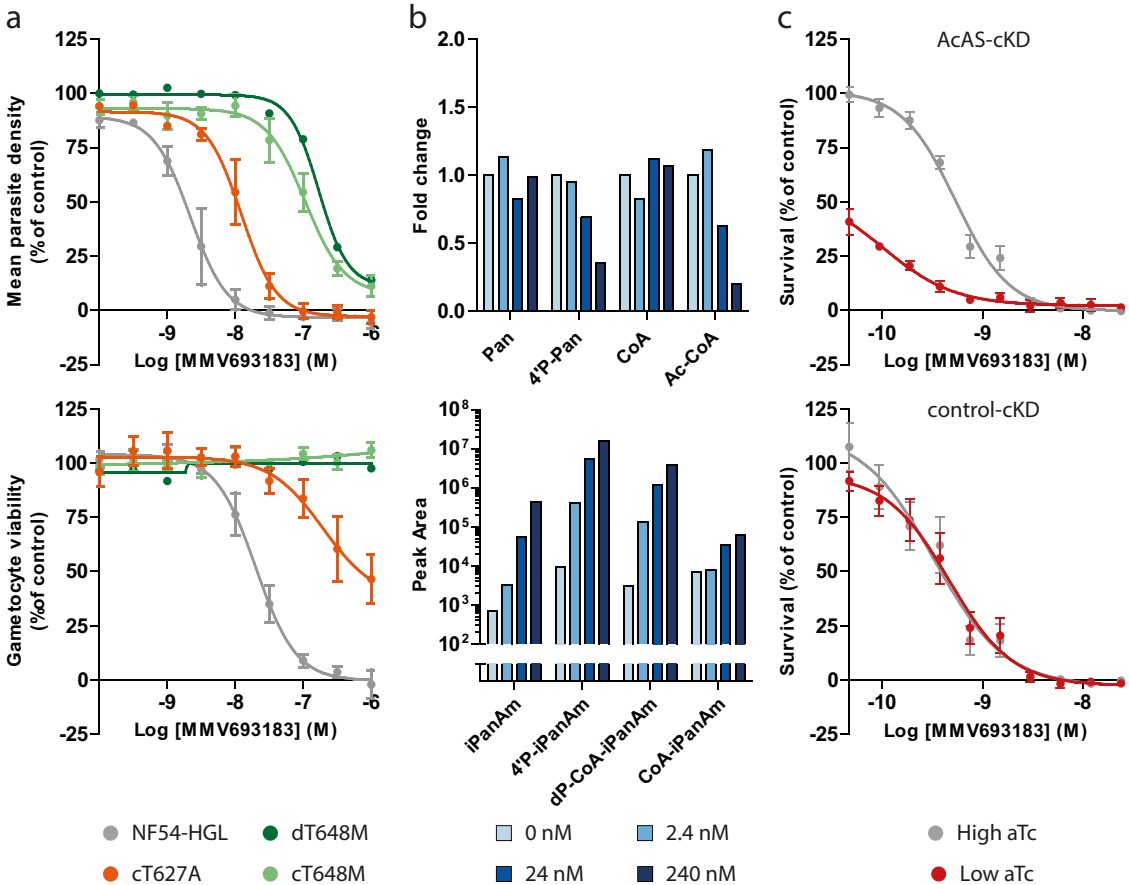

**Fig. 2 Role of AcAS in the mode of action of MMV693183. a** Drug-sensitivity profiles with asexual (upper panel) or sexual (lower panel) blood-stage parasites without a mutation (NF54-HGL) or with a T648M or T627A mutation in AcAS. An MMV693183-selected resistant parasite line (dT648M) was tested in one experiment with two technical replicates and the CRISPR-engineered parasites (cT648M and cT627A) were tested in three independent experiments (two technical replicates per experiment). The average value for mean parasite density relative to controls ± SEM are shown. **b** Concentration-dependent changes in levels of endogenous metabolites (upper panel) and pantothenamide antimetabolites (lower panel) upon treating *P. falciparum*-infected RBCs with MMV693183 or no drug. 3D7 parasites were synchronized at the trophozoite stage and treated with increasing concentrations of compound for 2.5 h and (anti)metabolites were quantified in two independent experiments with three technical replicates. Untreated parasites represent the background levels of MMV693183 metabolites. CoA could not be identified in the second experiment, therefore, only data from the first experiment are shown for the CoA level. The fold change is determined relative to no drug control (0 nM). Pan: pantothenate; 4'P-Pan: 4'-phosphopantothenate; Ac-CoA: acetyl-CoA. **c** Drug-sensitivity assays on conditional knockdown parasites of AcAS (upper panel) or a control target (lower panel) on asexual blood stages at low or high aTc were tested in three independent experiments ($N = 3$). The graphs show parasite survival based on a luminescence readout compared to controls ± SEM. aTc, anhydrotetracycline. Source data are provided as Source Data file.

mutant AcAS. 4'P-MMV693183 only showed poor or no activity against wild-type or mutant AcAS, respectively (Fig. 3b). This shows for the first time that the CoA-PanAm is the active metabolite that inhibits AcAS.

AcAS is predicted to provide acetyl-CoA for a variety of processes in the parasite, including fatty acid elongation in the endoplasmic reticulum and post-translational modifications in the cytosol and nucleus[27,32–35]. To begin to explore whether inhibition of AcAS could affect these downstream pathways, we studied the localization of AcAS using an endogenous GFP-tagged AcAS parasite line (Supplementary Fig. 12a-b). AcAS-GFP demonstrated a widespread, undefined intra-parasitic localization, although it was unclear whether it is also expressed in the nucleus in *P. falciparum* (Fig. 3c), as previously observed in apicomplexan parasites[27,32,36,37]. We also stained wild-type parasites with AcAS immune serum. A possible perinuclear and/or cytoplasmic signal was observed consistent with Prata et al.[35], but we could not detect an evident nuclear signal (Supplementary Fig. 13), which was observed recently[27,35].

**Pharmacokinetic properties**. Pharmacokinetic (PK) studies were performed in order to support a human dose prediction. The MMV693183 PK profiles in mice, rats, and dogs were examined using two-compartment models that were fit to plasma concentration-time data observed after oral and intravenous dosing (Supplementary Fig. 14, Supplementary Tables 7–9). Even though a Caco-2-permeability assay suggested moderate absorption and active efflux in vitro (Supplementary Table 10), MMV693183 was absorbed rapidly ($T_{max}$ of 0.5 h) in vivo and had an excellent oral bioavailability (64% in dogs to 121% in rats) (Supplementary Tables 7–9). The total clearance was 13.4, 21.1, and 11.5 ml/min/kg, and the half-life was 1.2, 3.1, and 4.2 h in mice, rats and dogs, respectively (Supplementary Tables 7–9). The plasma protein binding was overall low (ranging from 33% in mouse to 52% in human plasma) (Supplementary Table 11) and the blood to plasma ratio ranged from 0.91 to 0.98. The main route of metabolism in an in vitro human hepatocyte relay assay was an oxidation, followed by a glucuronide conjugation and dehydrogenation of the hydroxyl groups (Supplementary Table 12). An in vitro CYP reaction phenotyping assay using

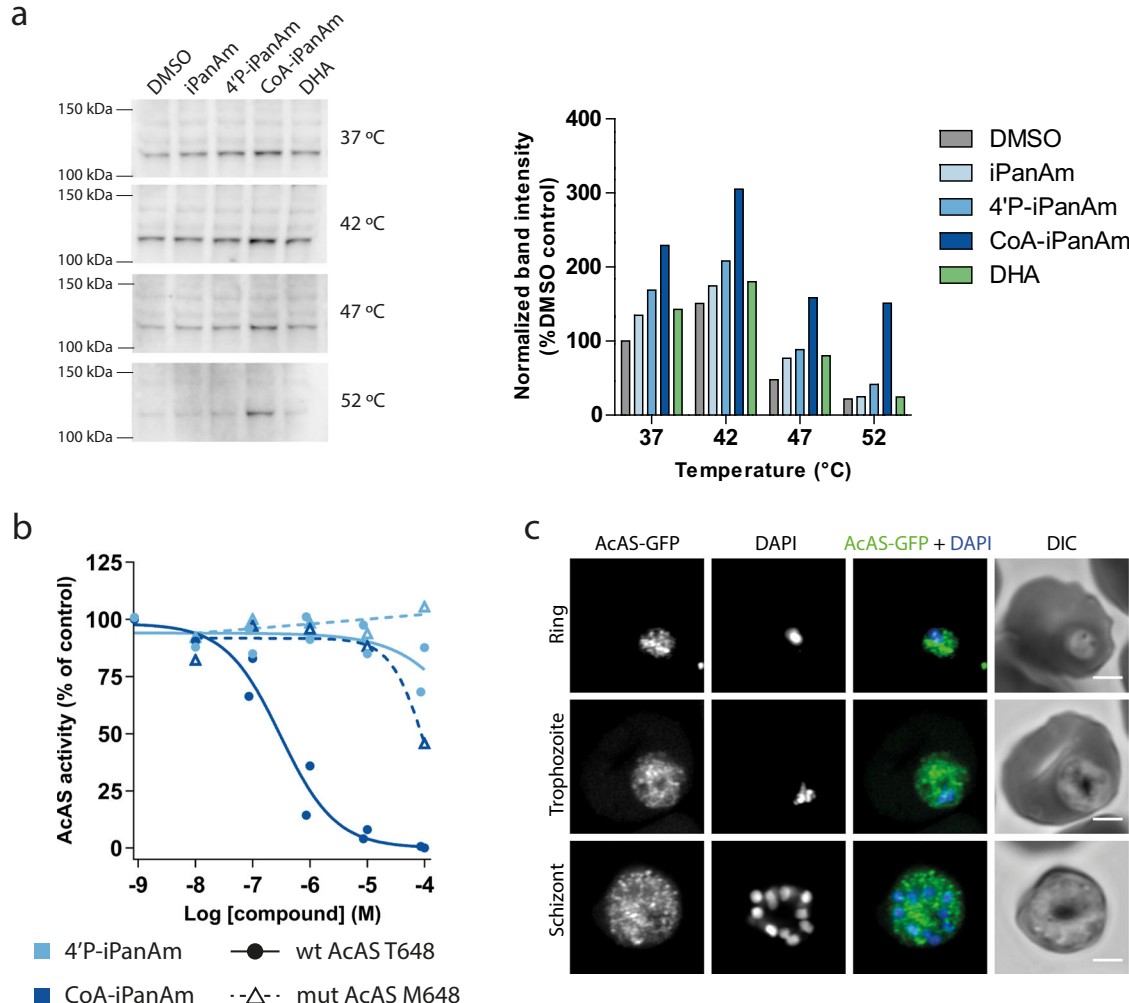

**Fig. 3 CoA-PanAm binds to and inhibits AcAS. a** Cellular thermal shift assay on *P. falciparum* lysate. Parasite lysate at 2.1 mg/ml was aliquoted and treated with 1 μM MMV693183 or metabolites thereof, or with DHA (negative control) for 30 min, followed by a 3-min incubation at different temperatures (*N* = 1). Protein stabilization was analyzed on a western blot (left panel) and band intensities (AcAS molecular weight = 113.8 kDa) were quantified and normalized to DMSO treatment at 37 °C (right panel). **b**, AcAS activity in a dose-response assay. Wild-type (T648) or mutant (M648) AcAS was immunopurified from NF54 or NF54-HGL and cT648M mutant parasite lysate, respectively, using rabbit immune serum. The activity was measured using $^{14}$C-labeled sodium acetate upon treatment with metabolites of MMV693183 and normalized to the no drug control (*N* = 4 for wild-type AcAS, *N* = 2 for mutant AcAS). **c**, Immunofluorescence microscopy of parasites with AcAS fused to GFP. Depicted are representative images of asexual blood-stage parasites stained with anti-GFP antibodies and DNA stained with DAPI. Scale bars, 2 μm. PanAm: MMV693183; 4'P-PanAm: 4'P-MMV693183; CoA-PanAm: CoA-MMV693183. Source data are provided as Source Data file.

human recombinant enzymes showed that CYP3A4, CYP3A5 and, to a lower extent, CYP2C19 were responsible for the oxidative metabolism of MMV693183 (Supplementary Fig. 15). In plasma and urine collected from dogs, we also detected oxidized, dehydrogenized and glucuronide-conjugated metabolites (Supplementary Table 13). In rats, 36–40% of the drug was eliminated in urine, while in dogs this was calculated to be 8.9%, excluding values from two dogs with only <20 ml urine (Supplementary Tables 14 and 15). Metabolic stability of MMV693183 was assessed in primary hepatocytes from mice, rats, and dogs. The observed values correlated well with the non-renal clearance observed in vivo, suggesting that the non-renal clearance is mainly via a hepatic route (Supplementary Table 16). For mice, no renal clearance data were available, but the total observed in vivo clearance amounted to 13.4 ml/min/kg whereas the predicted hepatic clearance was 7 ml/min/kg. This implies that 6.4 ml/min/kg (48%) of total clearance was contributed by the kidney, in line with the proportion of renal clearance in rats. In an

in vitro CYP induction assay performed in cryopreserved human hepatocytes from three donors, MMV693183 induced CYP3A4 at concentrations greater than 150 μM, and did not induce CYP1A2 nor CYP2B6 at concentrations up to 500 μM. A concentration-dependent increase in CYP3A4 mRNA was observed in 2 out of 3 donors, and the maximum fold induction observed was rather modest: 3.6-fold at 500 μM (Supplementary Table 17).

**Human pharmacokinetic predictions**. Two approaches were considered to predict human clearance. First, simple allometry predicted a clearance exponent of 0.967 (Supplementary Fig. 16), which was higher than typical ranges for this parameter (0.67–0.75)[38]. Consequently, a maximum life-span potential (MLP) correction was implemented resulting in an estimated total clearance of 1.8 ml/min/kg[39]. Second, human clearance was estimated from in vitro hepatocyte clearance assays. In a panel of cells from four different human donors, clearance ranged from

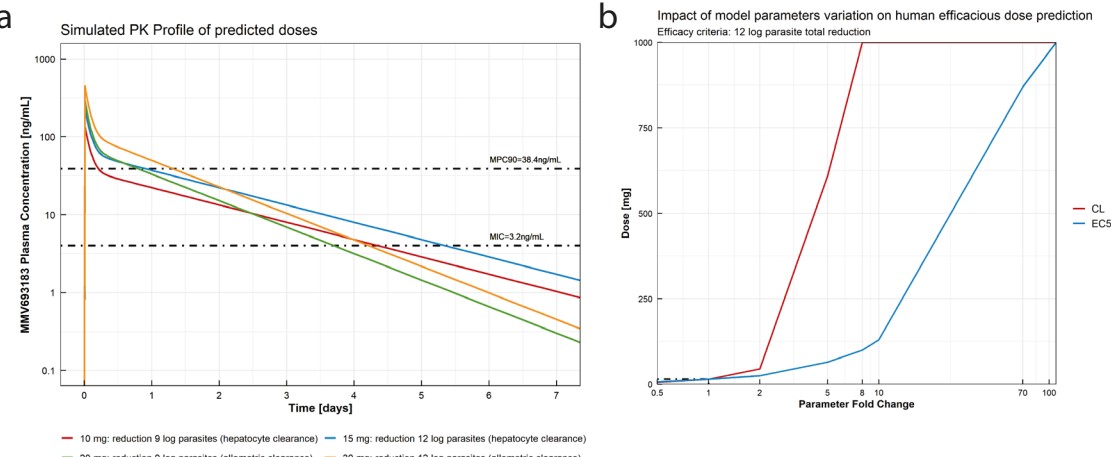

**Fig. 4 Human efficacious dose prediction. a** MMV693183 plasma concentration after the predicted efficacious human doses of 10, 15, 20, and 30 mg according to the efficacy criteria and human clearance prediction method. **b** Local sensitivity analysis of the impact of total clearance and $EC_{50}$ variation on the estimated efficacious dose, defined by a 12 log total parasite reduction efficacy criteria based on the prediction with the in vitro hepatocyte clearance assay.

0.07 to 0.4 µl/min/$10^6$ cells (Supplementary Table 18). Using the highest value (worst-case scenario) human hepatic clearance was predicted at 0.51 ml/min/kg. Human renal clearance was predicted at 0.60 ml/min/kg (Supplementary Table 19) based on renal clearance in dogs corrected for plasma protein binding (52%; Supplementary Table 11) and kidney blood flow, which was previously shown to be a good predictor[40]. Given the excellent correlation between the in vitro and in vivo data for the animal studies (Supplementary Table 16), the in vitro hepatocytic clearance prediction of 0.51 ml/min/kg was preferred over the allometry-derived value to predict total human clearance. This yielded a total clearance of 1.11 ml/min/kg, and a predicted human half-life of 32.4 h. Further human PK parameters were predicted using allometric scaling (Supplementary Fig. 16). Based on the Caco-2 permeability and thermodynamic solubility data (Supplementary Tables 4 and 10), the bioavailability was predicted to be 96% (GastroPlus) in line with a class I drug according to the biopharmaceutical classification system of the US Food and Drug Administration[41]. The predicted human parameters are shown in Supplementary Table 19.

**Prediction of the human efficacious dose using a PKPD model.** In vivo efficacy data from three female NSG mice studies were pooled to evaluate the PK-pharmacodynamics (PD) relationship of MMV693183 and derive key PD parameters such as MIC and $MPC_{90}$[42]. For all single-dose groups, the concentration of MMV693183 decreased to near or below the in vitro-determined $IC_{99}$ (36.1 nM) corrected for its free fraction (67% in mice) within 24 h (Supplementary Fig. 17), while 3D7 parasites were being cleared at 4–6 days. At later time points, these parasites recrudesced in all treatment groups (Supplementary Fig. 18). The PK profiles were well captured by a three-compartment PK model with zero-order absorption and a linear elimination (Supplementary Fig. 17 and Supplementary Table 20). PD parameters were estimated using an in vitro clearance model which is based on the $E_{max}$ model where the maximum killing rate was derived from the in vitro killing rate data while parasite clearance was estimated from the in vivo observations of parasitemia in time. The model describes efficacy as function of MMV693183 plasma concentration and showed an excellent fit to the in vivo data, including the recrudescence at later timepoints (Supplementary Fig. 18 and Supplementary Table 21). The combined PKPD

model predicted a MIC and $MPC_{90}$ of 3.2 and 38.4 ng/ml, respectively (Supplementary Table 22).

The human doses needed to achieve a 9 log total parasite reduction were predicted at 10 and 20 mg using the total clearance values from in vitro and allometric prediction, respectively (Fig. 4a). To achieve a 12 log total parasite reduction, the predicted doses were 15 and 30 mg, respectively (Fig. 4a, Supplementary Table 23). A local sensitivity analysis was performed on total clearance and $EC_{50}$ to evaluate the impact of the variation of both parameters on the human efficacious dose prediction. The most sensitive parameter for dose prediction was total clearance (Fig. 4b).

**MMV693183 safety.** Given that CoA metabolism clearly plays a central and crucial role in human cells, it was important to examine whether MMV693183, like MMV689258[15], acts selectively and specifically on the parasite without affecting the human host. Our studies revealed that treatment with MMV693183 was not cytotoxic to HepG2 cells. Since cell lines may be less metabolically active, we also examined the cytotoxicity in primary human or rat hepatocytes and did not find any cytotoxic effects (Supplementary Table 24). Furthermore, MMV693183 treatment did not affect human cardiac ion channels, including the Kv11.1 (hERG) channel, and MMV693183 was negative in AMES and micronucleus tests (Supplementary Table 24). In addition, MMV693183 did not show cross-reactivity to a panel of human receptors, enzymes, or channels. In this cross-reactivity assay, inhibition was <50% at a test concentration of 10 µM (Supplementary Table 24). A UV-scan did not reveal a liability for phototoxicity as there was no detectable absorption above 290 nm (Supplementary Table 24). Unlike the prophylactic antimalarial drug primaquine, MMV693183 did not show signs of hemolytic toxicity in a mouse model of human G6PD deficiency (Supplementary Fig. 19)[43].

Preliminary in vivo safety of MMV693183 was tested in rats with a seven-day repeat maximum tolerated dose study (MTD1) with 60, 200, and 600 mg/kg/day and a seven-day PK study at 60 and 200 mg/kg/day. In both studies body weight was unchanged, and no mortality occurred during the observation period, in any of the dose groups. Hematological and clinical chemistry parameters monitored in the MTD1 study showed no significant treatment-related deviations. Importantly, glucose, triglyceride, or urea concentrations, which were altered upon chemical

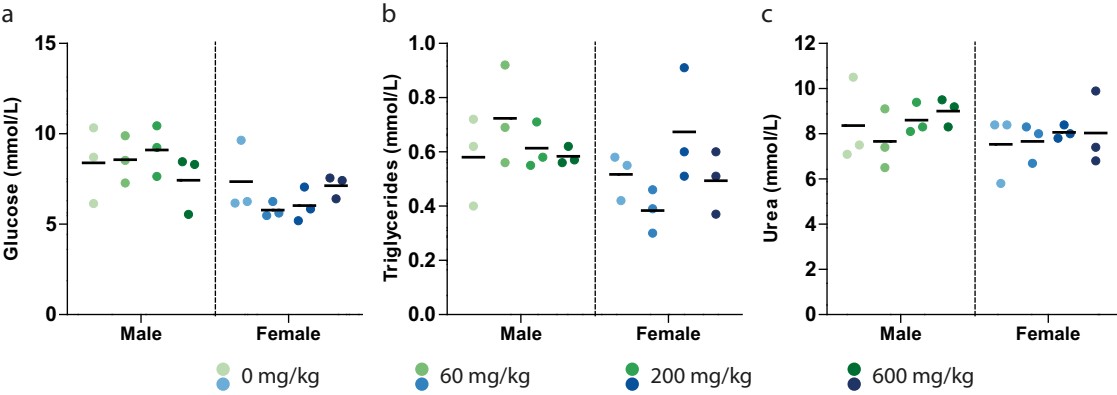

**Fig. 5 In vivo safety of MMV693183. a–c** Evidence of in vivo toxicity was examined in male and female Wister Han rats ($N = 3$ per condition) treated for seven days with MMV693183. Glucose (**a**), triglycerides (**b**) and urea (**c**) concentrations were measured in rats (male or female) treated with different doses of MMV693183. Significance was determined using One-Way ANOVAs with the Bonferroni's Multiple Comparison Test, but none were significant. Source data are provided as Source Data file.

disruption of the CoA pathway by hopantenate in mice in a previous study[44], were not affected by MMV693183 in any of the dose groups (Fig. 5a–c). Both the MTD1 and PK study identified the liver as the primary target organ of toxicity. Livers of male and female rats showed a dose-dependent increase in weight, up to 71% in males in the 600 mg/kg/day dose group in the MTD1 study (Supplementary Table 25). This observation was associated with histopathological liver changes consisting of centrilobular hepatocellular hypertrophy noted in all dosing groups, with a higher incidence and severity in males (Supplementary Tables 26 and 27). Furthermore, these changes were seen in conjunction with liver necrosis in 1 male out of 18 at 60 mg/kg/day and in 1 female out of 18 of each dose group, and with an increase in the number of hepatocellular mitoses in 1 female out of 18 of each dose group in the PK study. Minimal extramedullary hematopoiesis was also noted in most animals in this study (Supplementary Table 27). MMV693183 administration was associated with a major decrease (up to 11-fold) of exposure at later timepoints, starting from day 3 and up to day 7, in males at both dose levels and in females at the high dose only (Supplementary Tables 28 and 29). Based on these preliminary results a 3-day MTD 2 and an 8-day Dose Range Finding (DRF) study were performed to provide complementary information on the MMV693183 toxicity vis-à-vis of its exposure. In the MTD2 phase of the study no relevant toxicity was observed after three consecutive days of dosing at 300, 600, and 1000 mg/kg/day, respectively. Based on the adverse histological findings noted in the previous PK rat study, dose levels of 10 and 30 mg/kg/day were selected for the DRF phase of the study. Liver weight was unchanged (Supplementary Table 25), and there was an absence of liver microscopic findings and other test item-related adverse histological changes in both dosage groups and sexes in the DRF study. The administration of MMV693183 at 10 and 30 mg/kg/day was associated with a significant decrease (up to 4-fold) in systemic exposure on day 7 when compared to day 1, predominantly in males dosed at 30 mg/kg/day (Supplementary Tables 28 and 29). The cumulative exposure in the 30 mg/kg/day group was >30-fold above the most conservative prediction of human efficacious exposure (Supplementary Fig. 20), indicating a safety window in support of further (pre)clinical development of MMV693183.

## Discussion

Following an extensive chemical optimization process, we have identified the novel compound iPanAm MMV693183, which has low nanomolar potency against asexual blood stages of both *P. falciparum* and *P. vivax*, and against *P. falciparum* gametocytes. We showed its favorable physicochemical properties with the potential to be developed into a safe single-dose malaria cure. Furthermore, we revealed that the antimetabolite, CoA-MMV693183 inhibits AcAS, thereby targeting an unused pathway for antimalarial therapy. These promising characteristics of MMV693183 support the recent selection of this drug for continued (pre)clinical development[4].

While pantothenate analogs have long been explored, stable and highly potent pantothenamides against *P. falciparum* have only been developed in the last decade[9,12–14] and MMV693183 is the first pantothenamide to meet the criteria for further (pre)clinical development. This compound has improved in vitro and in vivo potency, metabolic stability, and a prolonged predicted human half-life compared to previously synthesized pantothenamides[13–15]. Its promising potency is reflected in the predicted human efficacious single dose of ≤ 30 mg for the treatment of clinical malaria. With this predicted efficacious dose, MMV693183 fits well within the portfolio of several antimalarials that all have the potential to act as a single dose cure with a dose range of 50 to 400 mg and an MPC ranging between 10 and 240 ng/ml[45–51]. Furthermore, MMV693183 is highly potent against *P. vivax*, another major contributor to the malaria burden[1], supporting the activity of PanAms against multiple species also including *P. knowlesi*[52]. The importance of combining MMV693183 with a partner drug is highlighted by the possibility of generating in vitro resistance against this compound that is compatible with transmission via the mosquito vector. The spread of resistance may also be affected by a possible fitness cost for resistant parasites, as was observed for the previously generated PanAm-resistant parasites with a mutation in AcAS and ACS11[15]. However, it is still unknown whether MMV693183-resistant parasites have a fitness defect. Fortunately, the T648M mutation has not been identified in > 2000 field isolates from 14 countries[53], suggesting that there is no pre-existing resistance. The recrudescence observed in our in vivo efficacy experiments is in line with the fast clearance of the compound in mice and well captured by the PKPD model presented. Nevertheless, we cannot exclude the possibility that T648M or other resistance mutations are selected in vivo, and this is the subject of our ongoing and future work, including anticipated human volunteer studies. Reassuringly, the minimum inoculum for resistance development against MMV693183 was $10^9$, which is considered to sufficiently reduce the risk of resistance against new antimalarials[54]. Further research to identify the ideal partner drug is needed. However, we

could envision a drug combination that is able to target pantothenamide-resistant gametocytes and prevent transmission of the resistance mutation.

The identified mutation in *AcAS*, but not in *ACS11*, in MMV693183-resistant parasites, supported our previous hypothesis that AcAS is the primary target of metabolized iPanAms[15,21,22]. Here, we present definitive proof that CoA-PanAm is the active metabolite that inhibits AcAS, while the inhibition of mutant AcAS is strongly reduced. This same phenotype has also been recently observed after inhibition by chemically different AcAS inhibitors[27], showing that AcAS is a good drug target. However, the $IC_{50}$ of AcAS inhibition in the biochemical assay was above the antiparasitic $IC_{50}$. This may be explained by (i) the instability of the CoA-PanAm leading to reduced concentrations in the enzyme assay; (ii) additional activity of dP-CoA-PanAm; (iii) accumulation of CoA-PanAm within the parasite reaching higher concentrations;[15] or (iv) a difference in concentration of CoA and/or acetate in the AcAS assay compared to physiological levels. With the previously suggested role of AcAS in regulating the acetylome, transcriptome and metabolome (including fatty acid elongation), the corresponding cytoplasmic/perinuclear and nuclear localization of AcAS, and the reduction in histone acetylation upon treatment with AcAS inhibitors[27,32,33,35–37,55], it could be hypothesized that these pathways are affected by CoA-PanAms. Similar to iPanAms, inhibitors of histone deacetylases or acetylases (HDAC or HAT), enzymes that regulate histone acetylation, have dual-stage activity targeting both asexual and sexual blood-stage parasites[56–58]. The marked difference between the female and male sexual-stage activity of MMV693183 could be related to one of the possible downstream consequences of AcAS inhibition that may be more important in female than in male gametocytes. Alternatively, the differential activity against male gametocytes compared to females may be explained by lower availability of iPanAm; for example, through reduced uptake, increased export, reduced subcellular accumulation at the site of activity, or increased breakdown. While the target of iPanAms has been identified, the further downstream consequences are not yet understood.

It is clear that PanAms target a central pathway of *Plasmodium* parasites, which is conserved among many eukaryotes and prokaryotes. Previous studies on hopantenate, a compound that affects CoA metabolism, showed lethal toxicity within 15 days, a significant reduction in glucose and altered liver metabolism in mice on a pantothenate-free diet[44]. It is therefore of utmost importance to test the safety of PanAms. MMV693183 did not show relevant activity against a large panel of human enzymes, receptors and channels. Even though no off-target activity was identified in our study, a cautionary note could be the weak effect on HDAC11. In a preliminary safety study, MMV693183 did not reduce glucose, and was not lethal to rats within the 7 day period. This was in contrast with hopantenate treatment, although these mice were on a pantothenate-free diet[44].The liver was identified as the primary target of toxicity upon MMV693183 treatment. At doses of 60 and 200 mg/kg/day for 7 days, liver hypertrophy was associated with reduced exposure at day 7 compared to day 1. This signature is commonly found with compounds that induce CYP expression in rats[59]. For MMV693183, such auto-induction is unlikely to happen in humans as our data indicated an absence of CYP induction at concentrations up to 150 μM whereas the $C_{max}$ at the predicted human efficacious dose was estimated at ~1 μM. At dose levels of 30 mg/kg/day for 7 days MMV693183 was well tolerated in rats providing a provisional exposure safety margin of >30-fold in comparison to the predicted human efficacious exposure (Supplementary Fig. 20). This provides a solid foundation for further assessment of safety during the preclinical

development of MMV693183, which will include studies in other rodents and dogs.

A few limitations of our PKPD model need to be considered, which could affect the final dose predictions. The PK model based on humanized mouse data showed a high relative standard error for a few of the estimated parameters, which may lead to uncertainties in the PD parameters used for the final dose predictions, such as the $EC_{50}$. However, a sensitivity analysis showed a limited impact of the variation of $EC_{50}$ on the dose predictions, identifying MMV693183 clearance in humans as the most sensitive parameter. Our prediction of human hepatic clearance values on basis of allometry gave a value that was ~3.5-fold higher than the value predicted from in vitro metabolism studies. Nevertheless, this worst-case scenario predicted a total human dose of 30 mg to achieve a 12-log reduction in parasitemia. This is encouraging but should be verified in future studies addressing human PK and efficacy.

In conclusion, we provide a new preclinical candidate MMV693183 that is a promising multi-stage active compound and that acts on a pathway that is not currently targeted by clinical antimalarials. This agent has the potential to be developed into a safe single-dose cure, and upon successful development may therefore aid in ongoing efforts to achieve malaria elimination.

## Methods

**Ethics statement**. Animal experiments performed at The Art of Discovery (TAD) were approved by The Art of Discovery Institutional Animal Care and Use Committee (TAD-IACUC). This committee is certified by the Biscay County Government (Bizkaiko Foru Aldundia, Basque Country, Spain) to evaluate animal research projects from Spanish institutions according to point 43.3 from Royal Decree 53/2013, from the 1st of February (BOE-A-2013-1337). All experiments were carried out in accordance with European Directive 2010/63/E. Mice were housed under 12/12 h light-dark cycle, 22 ± 2 °C, and 40–70% humidity.

The animal experiments carried out at the Swiss Tropical and Public Health Institute (Basel, Switzerland) were adhering to local and national regulations of laboratory animal welfare in Switzerland (awarded permission no. 2303). Protocols are regularly reviewed and revised following approval by the local authority (Veterinäramt Basel Stadt). Mice were housed under 12/12 h light-dark cycle, 21 ± 2 °C, and 40-70% humidity.

Aptuit is committed to the highest standards of animal welfare and is subject to legislation under the Italian Legislative Decree No. 26/2014 and European Directive No. 2010/63/UE. Animal facilities are authorized by the Italian Ministry of Health with authorization n. 23/2017-UT issued on 29th November 2017 according to art. 20 of Legislative Decree No. 26/2014. Furthermore, general procedures for animal care and housing are in accordance with the Association for Assessment and Accreditation of Laboratory Animal Care (AAALAC) recommendations. Dogs were housed under 12/12 h light-dark cycle, 19-21 °C, and 45–65% humidity.

Animal procedures to determine the hemolytic toxicity were approved by the University of Colorado Anschutz Medical Campus Institutional Animal Care and Use Committee. Mice were housed under 14/10 light-dark cycle, 72 ± 2 °F, and 40% ± 10% humidity.

All animal studies had the approval of the Institutional Animal Ethics Committee (IAEC) of TCG Lifesciences Pvt. Ltd and were conducted in accordance with the guidelines of the Committee for the Purpose of Control and Supervision of Experiments on Animals (CPCSEA), Government of India. Rats and mice were housed in groups under 12/12 h light-dark cycle, 22 ± 2 °C, 50 ± 20% humidity.

Rat toxicity studies were performed at Charles River Laboratories (France) in accordance with the ICH S5(R2) guideline requirements and the respective Institutional Animal Care and Use Committees for care and treatment of laboratory animals. All animals were housed under standard laboratory conditions that have been approved by the respective Institutional Animal Care and Use Committees for care and treatment of laboratory animals. The seven-day repeat (MTD1) dose study in the Netherlands was done was reviewed and agreed by the Animal Welfare Body of Charles River Laboratories Den Bosch B.V. within the project license AVD2360020172866 approved by the Central Authority for Scientific Procedures on Animals (CCD) as required by the Dutch Act on Animal Experimentation (December 2014). Rats were housed up to 5 animals of the same sex and treatment per cage at temperatures between 18 °C-25 °C, humidity ≥ 35% or at 40–70%, 12 h light and 12 h dark (except during designated procedures).

More information on animal experiments can be found in Supplementary Table 30.

For collection of blood for ex vivo activity studies in Brazil, Uganda, and Cambodia, all participants or their parents/guardians signed a written informed consent before blood collection. Patients were promptly treated for malaria after

blood collection, following national guidelines. The ex vivo activity study in Brazil was approved by the Ethics Committee from the Centro de Pesquisa em Medicina Tropical - CEPEM-Rondônia (CAAE 61442416.7.0000.0011). The ex vivo activity study in Tororo, Uganda was approved by the Makerere University Research and Ethics Committee, the Uganda National Council for Science and Technology, and the University of California, San Francisco Committee on Human Research. All isolates in the Cambodia study were collected during therapeutic efficacy studies (TES) upon protocol acceptance from Cambodian National Ethical Committees (NECHR-077, NECHR-087, NECHR-092 & NECHR-099).

The human biological samples were sourced ethically and their research use was in accord with the terms of the informed consents under an IRB/EC approved protocol.

**Chemistry.** The synthesis of the iPanAms is included in patent application EP3674288A1[24], which states: "Characteristic features of the analogs concern the moieties flanking the inverted amide; the carbon atom flanking the inverted amide in the center portion of the molecule could comprise a methyl substituent, the two nitrogen atoms are separated by a linker of two carbon atoms, and the moiety flanking the inverted amide at the distal portion of the molecule is a (hetero) aromatic, optionally substituted, ring or ring system, bonded directly to the carbonyl group of the inverted amide." Synthesis of the five new iPanAms are described in the Supplementary Methods.

Crystal screening for each further characterized pantothenamide was performed by a commercial service (Crystal Pharmatech Co., Ltd.) under 36 conditions using a variety of crystallization methods, including liquid vapor diffusion, slow evaporation, slurry conversion and salt/co-crystal formation and solvents. X-ray power diffraction (XRPD) patterns were collected by Bruker X-ray powder diffractometers.

**Parasite culture and in vitro efficacy of pantothenamides.** The *P. falciparum* strains Dd2-B2 (a clone of Dd2), 3D7, NF54 and the luminescent-reporter strain NF54-HGL[60] were cultured in RPMI 1640 medium supplemented with 25 mM HEPES, 382 μM hypoxanthine, 26 mM NaHCO$_3$, 10% human blood type A serum or 0.5% AlbuMAX II, and 3-7.5% human blood type O red blood cells (RBCs) (Sanquin, the Netherlands) at 37 °C in 3% O$_2$, 4% CO$_2$.

Replication assays were performed using a SYBR Green method as described previously[61]. Briefly, asynchronous parasites were diluted to 0.83% parasitemia, 3% hematocrit in 30 μl medium, and added to 30 μl of diluted compounds in medium (0.1% DMSO final concentration) in black 384-wells plates. After a 72-h incubation, 30 μl of SYBR Green diluted in lysis buffer was added according to the manufacturer's protocol (Life Technologies). Fluorescence intensity was measured on a BioTek Synergy 2 Plate Reader after 1-h incubation and was normalized to a DMSO control (100% growth) and DHA- or epoxomicin-treatment (no growth). To define the IC$_{50}$ of MMV693183-resistant Dd2-B2 parasites, ring-stage cultures at 0.3% parasitemia and 1% hematocrit were exposed for 72 h to a range of concentrations of MMV693183 along with drug-free controls. Parasite survival was assessed by flow cytometry on an Accuri C6 (BD Biosciences) with BD C6 Plus software using SYBR Green and MitoTracker Deep Red FM (Life Technologies) as nuclear stain and vital dye, respectively, and data were analyzed using FlowJo (10.5.0). To assess the effect of conditionally perturbing AcAS expression and treatment with MMV693183 on parasite growth, synchronous ring-stage AcAS conditional knockdown parasites (AcAS-cKD) or a control conditional knockdown line (control-cKD; previously generated parasites with a yellow fluorescent protein with regulatory TetR aptamers in the 3′untranslated region (UTR) integrated in the *cg6* chromosomal locus[28]) were cultured in high (500 nM) and low (1.5 or 0 nM, respectively) concentrations of anhydrotetracycline (aTc) and incubated with serially diluted MMV693183. Luminescence was measured after 72 h using the Renilla-Glo(R) Luciferase Assay System (Promega E2750) and the GlomAX® Discover Multimode Microplate Reader (Promega). The luminescence values were normalized to DMSO vehicle (100% growth) and 200 nM chloroquine-treated (no growth) samples as controls.

The stage-specific effect of MMV693183 against asexual blood stages was measured using a modification of a previously described method[62]. Asynchronous NF54 parasites were cultured in a semi-automated shaker system[63] and synchronized once with 5% D-sorbitol. To assess the ring-stage activity of MMV693183, this synchronized culture was treated again with sorbitol 31 h after the initial treatment. This yielded a parasite culture with ≥ 90% rings. To assess the schizont-stage activity of MMV693186, the initial synchronized culture was treated again with sorbitol 7 h later. Subsequently, parasites were cultured for another 17 h to yield a parasite culture of ≥ 90% early schizont stages. The ring and schizont synchronized cultures were diluted to 0.5% and then incubated with either 50 nM DHA or 50 nM MM693183 for 24 h in duplicate. Subsequently, the drug was washed out and parasites were cultured for another 24 h. Parasitemia was determined using Giemsa-stained thin smears at 0 h, 24 h, and 48 h after drug exposure.

The antimalarial killing rate was determined by GlaxoSmithKline (GSK, Tres Cantos, Madrid, Spain) as described previously[26]. Briefly, 0.5% 3D7 (BEI Resources) *P. falciparum* parasites (≥80% ring-stage population) at 2% hematocrit were treated with 10x IC$_{50}$ of MMV693183 (40 nM in 3D7 parasites) or pyrimethamine (0.94 μM) for 120 h and the drug was renewed daily. Parasite samples were taken every 24 h and drug was washed out, followed by four independent, 3-fold serial dilutions in 96-wells plates. The number of viable

parasites was determined on day 21 and 28 by counting the wells with parasite growth. Parasite growth was measured by uptake of $^3$H-hypoxanthine in a 72-h assay and was back-calculated to viable parasites using the following equation $X^{n-1}$ where n is the number of parasite-positive wells and X the dilution factor.

Gametocyte viability assays on NF54-HGL parasites were performed using an adapted high-throughput protocol as previously described[15,64]. In short, asexual blood-stage parasite cultures were set up at 1% parasitemia in a semi-automated shaker system at 5% hematocrit[63]. From day four until day eight or nine, parasites were treated with 50 mM *N*-acetyl glucosamine to eliminate all asexual blood-stage parasites. Subsequently, gametocytes were isolated by a Percoll density gradient centrifugation[64]. At day 11, gametocytes were seeded (5,000 per well) in 30 μl in white 384-well plates containing 30 μl of compounds diluted in medium (0.1% DMSO). After a 72-h incubation, 30 μl of ONE-Glo reagent (Promega) was added according to manufacturer's protocol and luminescence was quantified using the BioTek Synergy 2 Plate reader. Values were normalized to DMSO- and epoxomicin or dihydroartemisinin-treated controls.

The activity of MMV693183 against female and male gametocytes was assessed in a dual gamete formation assay (DGFA) as described previously[65]. Briefly, mature *P. falciparum* NF54 gametocyte cultures were added to 384-well plates containing DMSO or different concentrations of MMV693183 (in < 0.25% DMSO) or Gentian Violet (12.5 μM). After a 48-h incubation, gamete formation was stimulated by a drop in temperature (from 37 °C to 26 °C), and the addition of xanthurenic acid (2.5 μM). At 20 min after induction, exflagellation was recorded by automated time-lapse microscopy. After data collection, the plate was returned to a 26 °C incubator and incubated for 24 h. Female gamete formation was assessed by live staining with a Cy3-labelled anti-Pfs25 monoclonal antibody (1:2222) and recorded by automated microscopy.

To assess parasite development in hepatocytes, cryopreserved human primary hepatocytes (Tebu-Bio lot: HC10-10) were thawed according to the manufacturer's protocol and seeded (50,000 cells per well) in collagen-coated 96-well plates (Greiner). Cells were cultured at 37 °C in 5% CO$_2$ and the medium was refreshed after 3 h and 24 h. Salivary glands from *Anopheles stephensi* mosquitoes were dissected to obtain NF54 sporozoites that were added (60,000 per well) to hepatocytes 48 h post-thawing. Plates were spun down and sporozoites were incubated with hepatocytes for 3 h. Subsequently, sporozoites were aspirated and compounds diluted in hepatocyte medium, were added to the hepatocytes (0.1% DMSO final concentration). Medium-containing compounds were refreshed daily for four days. Hepatocytes were fixed with ice-cold methanol and samples were blocked with 10% fetal bovine serum (FBS) in PBS. Samples were incubated with rabbit anti-HSP70 (1:75, StressMarq) in 10% FBS for 1-2 h followed by incubation with secondary goat anti-rabbit AlexaFluor 594 antibody (1:1000, Invitrogen) in 10% FBS for 30 min. Samples were washed with PBS containing 0.05% Tween 20 between different steps. Cells were imaged on the Biotek Cytation and images were analyzed automatically using FIJI software.

**In vivo efficacy of pantothenamides.** The effect of pantothenamides on *P. falciparum* Pf3D7$^{0087/N9}$ [66] in vivo was assessed in female NSG mice (NODsci-dIL2Rγ$^{null}$) at the Swiss Tropical and Public Health Institute (Basel, Switzerland) as described previously (Supplementary Table 30)[15,67]. Briefly, humanized mice were engrafted daily with human erythrocyte suspensions from days -11 to day 6. After 11 days (day 0), mice were injected intravenously with 3×10$^7$ infected RBCs in a volume of 0.1 ml. On day 4, groups of $n = 2$ mice were treated with a single dose pantothenamides or chloroquine (50 mg/kg) by oral gavage. The hematocrit of all dosed mice and an untreated control group ($n = 4$ mice) was determined by fluorescence-activated cell sorting and parasitemia was analyzed by microscopy on >10,000 RBCs as described before[68]. Samples to quantify compound metabolites were collected and prepared at different time points (1, 2, 4, 6, and 24 h after treatment) for each mouse by mixing 20 μl of whole blood with 20 μl of Milli-Q, followed by immediate freezing of samples on dry ice. For the preparation of CAL and QC samples, MMV693183 was dissolved in acetonitrile/dimethylsulfoxide (1/1, v/v) to a concentration of 1.00 mg/mL. Serial dilutions were prepared in the same solvent to concentrations 50 times higher than the corresponding CAL and QC concentrations. The concentrations were calculated under consideration of purity and salt factor. The spiking of the CAL and QC sample were performed at room temperature. The eight calibration levels were 5.00, 10.0, 50.0, 100, 500, 1000, 3750, and 5000 ng/mL and three QC levels 15.0, 100, and 3750 ng/mL were analyzed in duplicate, together with the study samples.

For the sample precipitation, to an aliquot of 10 μL mouse blood/water (1/1) study sample, 20 μL of acetonitrile containing the internal standard MMV1542001 at a concentration of 100 ng/mL was added. After vortex mixing, the samples were centrifuged for 10 min at 50,000 g at 8 °C. Subsequently, an aliquot of 20 μL of the supernatant was transferred to an autosampler vial and an aliquot of 5 μL was injected onto the LC-MS/MS system.

The quantification of MMV693183 was performed by column separation with reverse phase chromatography followed by detection with triple stage quadrupole MS/MS in the selected reaction monitoring mode. The samples were handled by an autosampler CTC PAL (CTC Analytics AG, Zwingen, Switzerland) set at 8 °C. The chromatography column used was a YMC Hydrosphere C18, 2.1 × 33 mm, 3 μm (YMC Co. Ltd., Kyoto, Japan), kept at room temperature. The aqueous mobile phase consisted of water containing 0.1 % formic acid, whereas the organic mobile

phase consisted of acetonitrile containing 0.1 % formic acid, pumped by HPLC pump from Agilent 1200 series (Agilent Technologies Inc, Santa Clara, CA, USA). The HPLC gradient was as follows: 95% phase A, 5% phase B for 0.20 min, followed by transition to 5% phase A, 95% phase B in 1 min, 2 min at these concentrations, and a subsequent increase to 95% phase A, 5% phase B in 0.05 min which is kept for the remaining 0.75 min.

The detection of MMV693183 was performed using a TSQ Access mass spectrometer (Thermo Fisher Scientific, San Jose, CA, USA) by selected reaction monitoring. The ion source was heated electrospray ionization, using a negative polarity. Following parameters were used: Voltage 2500 [V], Vaporizer temperature 350 [°C], Sheath gas 60 [au], Auxiliary gas 5 [au], Capillary temperature 350 [°C] and Collision gas pressure 1.0 [mTorr]. The transition used for the detection of MMV693183 was: parent ion 361.080 m/z—daughter ion 285.000 m/z. The transition used for the internal standard was: parent ion 325.030 m/z—daughter ion 195.000 m/z.

The concentration of the analyte was calculated using the internal standardization method. The area ratio of analyte to internal standard against the concentration of calibration samples was used for quantification. The acquisition and processing of data were performed using LCquan 2.5.6 and Xcalibur 2.0.7. Microsoft Office Excel 2007 Prof was used for calculations and statistical evaluation of data. For MMV693183 the fitting of data was done with a weighting factor of 1/X using a quadratic regression with the method of least squares. The Lower Limit of Quantification was 5.00 ng/mL in mouse plasma, whereas The Upper Limit of Quantification was 5000 ng/mL in mouse blood/water (1/1).

**Transmission-blocking activity**. The transmission-blocking activity of compounds was determined as described previously[69]. Briefly, NF54-HGL parasites were set up at 1% parasitemia in a semi-automated shaker system at 5% hematocrit. After 14 days of culturing, stage V gametocytes were treated with a range of concentrations of the compound for 24 h before feeding, or with 1 µM MMV693183 or 100 nM atovaquone directly upon feeding to *A. stephensi*. Eight days after the feed, luminescence was quantified to determine oocyst intensity.

**Ex vivo efficacy of pantothenamides**. For ex vivo pantothenamide activity studies, patients with *P. falciparum* or *P. vivax* were recruited at the Research Center for Tropical Medicine of Rondonia (CEPEM) in Porto Velho (Brazilian Western Amazon). A schizont maturation assay was performed using parasites obtained from mono-infected patients. A total of 44 patients were recruited who did not use any antimalarial in the previous months and/or present with symptoms of malaria, but had a parasitemia between 2,000 and 80,000 parasites/µl. Isolates from patients were excluded if (i) <70% of parasites were rings at the time of sample collection (*n* = 11), (ii) no schizont maturation was observed (*n* = 9), or if (iii) the number of inviable parasites in untreated control was higher than the number of maturated schizonts in the treated condition (*n* = 4), leaving 20 patients to be included. Peripheral venous blood (5 ml) was collected by venipuncture in heparin-containing tubes, plasma and the buffy coat were removed, RBCs were washed and subsequently filtered in a CF11 cellulose column. Blood was diluted to 2% hematocrit in either RPMI 1640 medium (*P. falciparum*) or McCoy's 5 A medium (*P. vivax*) supplemented with 20% compatible human serum. Parasites were incubated with MMV693183 at final concentrations ranging between 0.25 and 500 nM in a hypoxia incubator chamber (5% $O_2$, 5% $CO_2$, 90% $N_2$). The incubation of parasites with the compound was stopped when 40% of the ring-stage parasites reached the schizont stage (at least three distinct nuclei per parasite) in the untreated control wells. The number of schizonts per 200 asexual blood-stage parasites was determined and normalized to control. An assay was considered valid when the compound was incubated with parasites for at least 40 h.

An ex vivo growth inhibition assay was performed on fresh clinical *P. falciparum* isolates in Uganda. Blood was collected from patients aged ≥6 months presenting to the Tororo District Hospital, Tororo District, or Masafu General Hospital, Busia District with clinical symptoms suggestive of malaria, Giemsa-stained thick smears positive for *P. falciparum* infection, and ≥0.3% parasitemia determined by Giemsa-stained thin smears. Up to 5 ml of blood was drawn by venipuncture from 109 participants at Tororo District Hospital and 121 at Masafu General Hospital. Parasites were diluted to 0.2% parasitemia in 2% hematocrit and incubated for 72 h with serial dilutions of MMV693183 (0.1% DMSO) in a 96-well microplate and stored in a humidified modular incubator (2% $O_2$, 3% $CO_2$, 95% $N_2$). Parasite density was quantified by fluorescence after incubation with SYBR Green lysis buffer measured on a BMG Fluostar Optima plate reader, as previously described[70].

Field isolates were collected from patients infected with *P. falciparum* between 2017 and 2019 in western Cambodia (Kampong Speu) and eastern Cambodia (Kratie, Mondulkiri, Rattanakiri, and Pursat). Venous blood (5 ml) was collected by venipuncture, white blood cells were removed, and parasites were adapted to in vitro culture under 5% $O_2$ and 5% $CO_2$ in 2% $O^+$ human red blood cells (NBTCC, Phnom Penh) with RPMI1640, 0.2 mM hypoxanthine, 0.5% Albumax II and 2.5% human serum (NBTCC, Phnom Penh). In vitro drug susceptibility was measured on early ring stages (0–3 h postinvasion) diluted to 3% parasitemia and incubated for 72 h with serial dilutions of MMV693183 (0.05% DMSO) in a 384-wells plate at 0.01% hematocrit. Parasites were then fixed with 0.44% glutaraldehyde (#G5882, Sigma-Aldrich) for 15 min, permeabilized with 3% Triton (Sigma-Aldrich) for 10 min, and stained with 80 nM YOYO™-1 Iodide (#Y3601,

Invitrogen) for 45 min at room temperature in the dark. Endpoint readout to determine parasites density was performed with a High-Content Confocal Imaging microscope (Lionheart™ FX Automated Microscope, Biotek), $IC_{50}$ values were obtained using ICestimator software (http://www.antimalarial-icestimator.net/runregression1.2.htm).

**In vitro safety and toxicity assays**. Safety studies were performed by commercial services using their standard protocols. Off-target activities of 10 µM MMV693183 were investigated using binding, enzyme, and uptake assays (Eurofins CEREP, Celle-Lévescault, France). Phototoxicity of MMV693183 was assessed by exposing the compound to different wavelengths. Genotoxicity was investigated using the Ames test (Bacterial Reverse Mutation Assay) (Covance Laboratories Ltd, North Yorkshire, England). In vitro mammalian cell micronucleus screening assay was studied in human peripheral blood lymphocytes (BioReliance Corporation, Rockville, USA). Cardiotoxicity was determined against the hERG channel in an automated patch clamp assay using the Qpatch or against the $hNa_V$ 1.5, $hK_V$ 1.5 and $hCa_V$ 1.2 using a manual patch-clamp technique (Metrion Biosciences, Cambridge, UK). Cytotoxicity in human and rat primary hepatocytes was assessed using the CellTiter-Blue assay performed by commercial services (KaLy-Cell). The viability of HepG2 cells was monitored through the addition of 1 mM resazurin. Following 6 h incubation, the relative amount of reduced resazurin was detected in a fluorimeter and compared to the assay controls (vehicle: 0.1% DMSO and positive control: 10 µM puromycin).

**Exploratory in vivo safety and toxicology studies**. Hemolytic toxicity was determined in female NSG mice (Jackson Laboratories) using erythrocytes from a G6PD-A-deficient blood donor (0.4 u/g hemoglobin). Mice were engrafted with $3.5 \times 10^9$ human RBCs intraperitoneally for fourteen days to obtain >60% human RBCs. Mice were treated for four days with vehicle control (PBS) or MMV693183 (10, 25, 50 mg/kg), or primaquine (12.5 mg/kg) for three days. Spleen weight was quantified on day seven and hemolysis was assessed on day zero, four, and seven, by quantifying human RBCs and murine reticulocytes on a CytoFlex S Flow Cytometer (Beckman Coulter, CytExpert 2.3 software) using anti-glycophorin A-FITC (1:100) and anti-CD71-FITC (1:200) and anti-TER119-PE (1:400), respectively, and data were analyzed using FlowJo (version 10.7.1) (Supplementary Fig. 21).

Four non-GLP rat toxicity studies were performed for MMV693183: (1) a 7-day Maximum Tolerated Dose 1 (MTD1) study (study ref 20154166); (2) a 7-day Pharmacokinetic (PK) study (study ref 20223355); (3) a 3-day MTD2 study (study ref 20223357); and (4) an 8-day Dose Range Finding (DRF) study (study ref 20223357). See the summary of the study design in Supplementary Table 31. Nine-week-old Wistar (Crl:WI (Han)) rats were obtained from Charles River Laboratories (Domaine des Oncins, Saint-Germain-Nuelles, France). The Test Item (MMV693183) was administered by oral gavage at different doses and dosing regiments (Supplementary Table 30). Animals were examined daily for any altered clinical signs, body weights, and food consumption. Blood samples were taken at different timepoints after the test item administration, and toxicokinetic exposure assessed under the defined experimental conditions and analyzed using Phoenix (version 6.4 and 1.4) (Supplementary Table 32). Measurements included hematology and clinical chemistry measurements. Animals were euthanized and necropsied at the end of the dosing period. At necropsy, major organ systems and tissues were collected and weighed and then examined for gross lesions as well as for microscopic changes.

**In vitro metabolism, permeability, and protein binding**. Stability of MMV693183 in dog, rat, and hepatocytes, Caco-2 permeability, plasma protein binding, and blood to plasma ratio were analyzed through a commercial service (TCG Lifesciences). The stability of MMV693183 in human hepatocytes was quantified in a relay assay[25]. Briefly, cryopreserved human primary hepatocytes were thawed and cultured as described above. Hepatocytes were incubated with compounds in hepatocyte medium (0.1% DMSO) 24 h post-seeding. The supernatant was collected 1, 6, or 24 h after the addition of compound, spun down and the supernatant was stored at -80 °C. Supernatant from the 24 h treatment was pooled and transferred to a new hepatocyte plate seeded 24 h earlier. This process was continued until the 72-h incubation time was reached. Samples were analyzed on the LC-MS/MS system Thermo Scientific™ Vanquish™ UHPLC system or Thermo Scientific™ Q Exactive™ Focus Orbitrap with a HESI-II electrospray source in positive mode using a Luna Omega Polar C18, 50 ×2.1 mm, 1.6 µm column. The chromatography was performed at a flow rate of 0.8 ml/min using an 8.70-minute gradient to 70% Mobile Phase B (0.1% formic acid in methanol) and 30% Mobile Phase A (0.1% formic acid in MilliQ water), followed by 0.40-min gradient to 99% Mobile Phase B, and back to 1% Mobile Phase B Mobile in 0.30 min.

In vitro CYP reaction phenotyping was performed at Cyprotex (Alderly Park, United Kingdom). Briefly, MMV693183 (1 µM) was incubated with human CYP1A2, CYP2B6, CYP2C8, CYP2C9, CYP2C19, CYP2D6, CYP3A4, and CYP3A5 recombinant isoforms (Bactosomes™), at 5-time points over the course of a 45 min experiment. The remaining test compound at each time point was analyzed by LC-MS/MS.

The potential of MMV693183 to be an inducer of CYP1A2, CYP2B6 and CYP3A4 were investigated in vitro at Cyprotex (Alderly Park, United Kingdom),

using an mRNA endpoint across three donors in cryopreserved human hepatocytes (obtained from Lonza, Walkersville, USA). Cells were dosed with MMV693183 (1.5, 5, 15, 50, 150 and 500 μM for donor 1 and 5, 15, 50, 150, 250, and 500 μM for donor 2 and donor 3) for 72 h. Positive control inducers (omeprazole for CYP1A2, phenobarbital for CYP2B6, and rifampicin for CYP3A4), were incubated alongside the test compound. For mRNA assessment, cells were lysed and resultant samples analyzed using an ABI 7900 HT real-time PCR system. A cytotoxicity assessment using 3-(4,5-dimethyl-2-thiazolyl)-2,5-diphenyl-2H-tetrazolium bromide (MTT) was performed in one of the three donors used for the induction studies prior to dosing and demonstrated no change in cell viability or cell morphology changes over the concentration range tested.

**Pharmacokinetic properties**. The in vivo pharmacokinetic (PK) profiles of MMV693183 in blood and urine were determined in male CD1 mice, Sprague Dawley rats (both TCG Lifesciences), and Beagle dogs (Aptuit, Verona, Italy) (Supplementary Table 30). Briefly, mice and rats were dosed intravenously or by oral gavage with MMV693183 at 3 mg/kg or 30 mg/kg, respectively. A blood sample from the saphenous vein was collected into heparinized capillary tubes at multiple intervals between 0.25 and 24 h after dosing. Subsequently, samples were spun down at 1640 g for 5-10 min at 4 °C within 30 min after collection and plasma was collected. The PK study in dogs followed a cassette dosing of MMV693183, MMV693182 and MMV689258 with intravenous dosing of 1 mg/kg per compound and oral dosing of 2 mg/kg per compound. Urine was collected following an intravenous dose in both rat and dog, allowing renal clearance to be estimated. Human renal clearance was estimated from dog renal clearance as previously described[40] and other human PK parameters were predicted using allometric scaling and/or in vitro clearance data. PK parameters were calculated by naïve pooled approach using WinNonLin software (Phoenix, version 6.3).

**In vivo Pharmacokinetic-Pharmacodynamic (PKPD) relationship**. The effect of MMV693183 on *P. falciparum* Pf3D7[0087/N9] and Pf3D7[20161128TAD17N214] [66] was investigated in vivo using female NSG mice (Charles River) and was assessed in three different studies at TAD (The Art of Discovery, Spain) using their standard protocol (Supplementary Table 30)[67]. Briefly, humanized mice were engrafted daily with 0.7 ml of 50%-75% hematocrit human erythrocyte suspension until the end of the drug administration period to obtain a minimum of 40% human erythrocytes in peripheral blood during the entire experiment. Subsequently, mice were injected intravenously with $35.1 \times 10^6$ infected RBCs in a volume of 0.3 ml. When parasitemia reached 1% (three days after infection), mice were left untreated or were treated with DHA for four days (8 mg/kg) or with MMV693183 in different dose groups described in Supplementary Table 30. When parasitemia reached the lower limit of quantitation (<0.01%) after treatment, the total circulating human RBCs were maintained by injection of human RBCs every three to four days. Parasitemia was regularly quantified (every 24 to 72 h) in each mouse by staining 2 μl of tail blood and measured on the Attune NxT Acoustic Focusing Flow Cytometer (InvitroGen) as previously described[67]. The experimental designs are summarized in Supplementary Table 30. Samples to quantify MMV693183 were collected and prepared at different time points depending on the study (range 0.5 to 103 h after treatment) for each mouse by mixing 25 μl of whole blood with 25 μl of Milli-Q, followed by immediate freezing of samples on dry ice. Samples were processed under liquid-liquid extraction methods and analyzed by LC-MS/MS for quantitation in a Waters UPLC-TQD (Micromass, Manchester, UK). The lower limit of quantification for MMV693183 ranged from 1 to 5 ng/ml, depending on the study.

Data preparation, exploration and model pre- and postprocessing were performed using R (version 3.6.3) and R package IQRtools (version 1.2.1 IntiQuan GmbH). Non-linear mixed-effects (NLME) modeling was used to estimate the PK and PD parameters using Monolix (Lixoft version MLX2018R2). The population PKPD model was developed with a two-stage approach: first, a population PK model was determined; then, the individual PK parameters were used as regression parameters and the PD parameters were estimated. The PD model consists of the balance between a parasite net growth rate and a drug killing rate. The effect of MMV693183 concentration on the killing rate was estimated using an in vitro clearance model which is based on an $E_{max}$ model[71] with an additional clearance term to account for the removal of dead parasites from the body and where $E_{max}$ is fixed to the value derived from the in vitro parasite reduction rate (PRR) assay (Supplementary Fig. 22). The growth rate of log-transformed parasite concentration was fixed at 0.03 /h based on prior experiments but estimating growth rate interindividual variability. Different Hill coefficients were tested. The final model was selected based on model convergence, the plausibility of parameter estimates, visual inspection of observed and model-predicted time courses, standard goodness-of-fit plots and fit statistics such as Bayesian Information Criterion (BIC). The MIC and $MPC_{90}$ were calculated with the following formulas:

$$MIC = EC_{50} * \left( \frac{GR}{E_{max} + GR} \right)^{1/Hill} \quad (1)$$

$$MPC_{90} = EC_{50} * \left( 9 * \frac{CL_{para}}{CL_{para} + E_{max}} \right)^{1/Hill} \quad (2)$$

GR, parasite growth rate (1/h); MIC, minimum inhibitory concentration (ng/ml);

$MPC_{90}$, minimum parasiticidal concentration when killing/clearance rate reaches 90% of its maximum (ng/ml); $EC_{50}$, effective concentration required to obtain 50% of maximum effect (ng/ml); Hill, Hill factor determining the steepness of the exposure-response curve; $CL_{para}$, parasitemia clearance rate (hour$^{-1}$).

**The human efficacious dose estimation**. The human efficacious dose estimation, defined as the dose able to achieve at least 9 to 12 log total parasite reduction, was predicted performing simulations using the predicted human PK parameters and the PD parameters estimated from the female NSG mice studies. Two sets of predicted human PK parameters were considered. The first set included human hepatic clearance estimated from in vitro hepatocyte clearance and human renal clearance estimated from dog pharmacokinetic data using allometry. The second set included the total human clearance estimated using allometry and was considered as the worst-case scenario due to the higher total clearance estimated using this approach.

**Induction of drug-resistant parasites**. Dd2-B2 and NF54-HGL parasites were exposed to suboptimal concentrations of MMV693183 to induce and select for drug-resistant parasites. Dd2-B2 parasites were set up at different parasite densities ranging from $10^7$–$10^9$ and exposed to 3.5-9 × IC$_{50}$ (Dd2-B2 mean IC$_{50}$ = 3.0 nM) in at least three independent experiments. Drug media was changed daily until cultures were cleared and every other day subsequently. If cultures did not clear (<0.09% parasitemia), then the concentration of the drug was ramped up. Recrudescence was monitored by flow cytometry on an Accuri C6 (BD Biosciences) using BD C6 Plus for 60 days using SYBR Green and MitoTracker Deep Red FM (Life Technologies) as nuclear stain and vital dye, respectively, and data were analyzed using FlowJo (Supplementary Fig. 23). Selections of Dd2 parasites at $1 \times 10^5$ cleared within the first five days. The first selection of Dd2 parasites at $1 \times 10^7$ survived the first week and began to expand after treatment with 15 nM, after which the selection pressure was increased to 22 nM. Parasites were cleared within three days. Selections at $1 \times 10^7$ and $1 \times 10^8$ Dd2 parasites with 10.5 nM experienced normal growth after one full cycle and drug concentration was ramped up to 14 nM. Cultures cleared over a period of 6 days. Selections at $1 \times 10^9$ were set up at 12 nM and concentration was ramped up until reaching 27 nM over a period of 30 days. Cultures were never fully cleared as a few healthy rings were regularly visible. NF54-HGL parasites were treated with 3 × IC$_{50}$ for two weeks in two independent experiments and cultures were monitored by luminescence readout on the BioTek Synergy 2 Plate reader using ONE-Glo reagent (Promega). Whole-genome sequencing was performed to test for single nucleotide polymorphisms (SNPs) or copy number variations.

**Generation of transfection plasmids**. The mutation found in selected drug-resistant parasites was verified by introducing the point mutations in AcAS (T648M and T627A) in NF54-HGL parasites using a CRISPR-Cas9 system as described previously[15]. The oligonucleotides for the guide RNA and the donor template were cloned into a pDC2-based plasmid containing a Cas9 and guide cassette, using the BbsI and EcoRI/AatII restriction sites, respectively. Donor DNA was amplified by (overlap-extension) PCR amplification from genomic *P. falciparum* DNA and oligonucleotides for the guide RNAs were ordered (Sigma-Aldrich). The correct sequence and integration of both inserts were confirmed by Sanger Sequencing.

AcAS and ACS11 were C-terminally tagged with GFP using the Selection-Linked Integration (SLI) system[72]. The C-terminal homology region was cloned into the plasmid using NotI/MluI restriction sites. The resulting vector was digested using EcoRV/BstZ17I restriction sites to insert the 3′ UTR. The final resulting vector contained an apicoplast-mCherry cassette that was not used in this study. Donor DNA was amplified by PCR amplification from genomic *P. falciparum* DNA. Primers are defined in Supplementary Table 33.

**Plasmodium falciparum transfections**. A DNA-loaded RBC protocol was used for transfection[73]. Briefly, 100 μg of plasmid was loaded into RBCs by electroporation (310 V, 950 μF), and a trophozoite culture was added to these transfected RBCs. One day after transfection, parasites were treated with 2.5 nM WR99210 (Jacobus Pharmaceutical) for five days and cultured until they recovered. For the generation of the mutation in *AcAS*, parasites were cloned by limiting dilution, and integration of the mutation was confirmed by Sanger sequencing (Supplementary Fig. 24). For the generation of AcAS- and ACS11-tagged mutants, parasites were treated with 400 μg/ml G418 (Sigma) until parasites recovered (about 15 days). Subsequently, parasites were sorted on a customized FACSAriaII (BD Biosciences) based on a previously published protocol[74] and data was collected using BD FACSDiva software (8.0.1). In brief, a diluted sample was passed through a 70 μm nozzle. Single cells were isolated by removing doublets or cell aggregates based on FSC-H/FSC-W and SSC-H/SSC-W dot plots and selecting GFP-positive events measured using a 515 nm long-pass filter. Next, 30 (AcAS-tagged) or 100 (ACS11-tagged) parasites were placed back into culture. The gating strategy, processed in FlowJo (10.8.1), is presented in Supplementary Fig. 25. Successful integration of the transfection plasmids and the absence of wild-type parasites were verified by PCR (Supplementary Fig. 12).

**Metabolomics assays**. *P. falciparum* metabolomics analysis was performed by LC/MS as previously described[75]. Briefly, parasite cultures were tightly synchronized at the ring stage one cycle prior to extraction. Trophozoite cultures at 5-10% parasitemia were purified to >90% parasitemia by magnetic purification. Parasites were counted using a hemocytometer, aliquoted to $1 \times 10^8$ cells per condition in 5 ml medium and then placed into an incubator for 1 h to allow them to reach a metabolically stable state. Following the recovery period, MMV693183 was added at 1×, 10× or 100× IC$_{50}$ value and compared to a no-drug control in triplicate. After the incubation period, media was aspirated and the remaining culture was washed with PBS, and quenched using 90% methanol containing 0.25 μM [$^{13}$C$_4$$^{15}$N]-aspartate. Blank processing samples were also quenched in the same manner to assess background metabolite levels. The samples were centrifuged, the supernatant was collected in a new tube, dried using a nitrogen gas drying rack and stored at -80 °C until they were run on the LC/MS platform.

Samples were resuspended in 1 μM chlorpropamide in 3% HPLC-grade methanol diluted in HPLC-grade water and run on a Thermo Exactive Plus Orbitrap HPLC/MS in negative mode with a scan range of 75-1000 m/z using a C18 Water Xselect HSS T3 column with 2.5 μm particle diameter. Chromatography was performed using a 25-min gradient of 3% methanol with 10 mM tributylamine and 15 mM acetic acid (solvent A) and 100% methanol (solvent B). For each analytical run, a pooled sample was generated by combining equivalent volumes of each parasite sample to assess metabolite detection and run at the beginning, middle, and end of each analytical batch to detect any possible time-dependent sensitivity changes. The full metabolomics datasets are publicly available on the Metabolomics Workbench database under ST001985[76].

**Whole genome sequencing**. The Dd2-B2 parent and resistant clones were subjected to whole-genome sequencing at the Columbia University Irving Medical Center using the Illumina Nextera DNA Flex library preparation protocol and NextSeq 550 sequencing platform. Briefly, 150 ng of genomic DNA was fragmented and tagmented using bead-linked transposomes and subsequently amplified by 5 cycles of PCR to add dual index adapter sequences to the DNA fragments. The libraries were quantified, pooled and sequenced on the Illumina NextSeq high output flow cell to obtain 150 bp paired-end reads.

The sequence data generated were aligned to the *P. falciparum* 3D7 genome (PlasmoDB version 36.0) using BWA (Burrow-Wheeler Alignment). PCR duplicates and reads that did not map to the reference genome were removed using Samtools and Picard. The reads were realigned around indels using Genome Analyses Tool Kit (GATK) RealignerTargetCreator and base quality scores were recalibrated using GATK Table-Recalibration. GATK HaplotypeCaller (Min Base quality score ≥ 20) was used to identify all possible variants in clones. Variants were filtered based on quality scores (variant quality as a function of depth QD > 1.5, mapping quality > 30) and read depth (≥ 5) to obtain high-quality SNPs that were annotated using snpEFF. The list of variants from the resistant clones were compared against the Dd2-B2 parent to obtain homozygous SNPs present exclusively in the resistant clones. Copy number variations were detected using the BicSeq package by comparing the read counts of the resistant clones against the Dd2-B2 parent. Integrative Genomics Viewer was used to verify the SNPs and copy number segments in the resistant clones.

Genomic DNA from the parental NF54 line and MMV693183-induced resistant lines were sequenced at the Pennsylvania State University according the Illumina® Truseq Sequencing protocol. Following sequencing, the data were processed using the Tadpole Galaxy scientific data analysis platform[77]. Briefly, the Trimmomatic tool was used to trim adapter sequences and genomes were mapped using the Map with BWA-MEM tool against a *P. falciparum* 3D7 reference genome. The Filter Sam or Bam, output Sam or Bam tool was used to consolidate the reads and generate the BAM file for the remaining analyses. The Depth of Coverage on Bam File tool was used to assess the depth of coverage of the dataset. Finally, the Freebayes – Bayesian genetic variant detector tool was used to assess the data for SNPs, inserts, and deletions.

**Generation of antigen for AcAS antibody production**. Recombinant protein fragment used for immunization was obtained by cloning the first 414 nucleotides of a codon-optimized coding sequence of AcAS in frame with an N-terminal Glutathione S-transferase (GST)tag, into the expression vector pGex-4T. The vector was transformed into competent *Escherichia coli* BL21 (DE3) expression cells to express a recombinant protein fragment. Protein production was induced in 250 ml log-phase growing cells with 500 μM IPTG for 3 h at 30 °C. After incubation, the cells were collected, resuspended in 10 ml 20 mM Tris-HCl (pH 7.5) and disrupted by sonicating 3 times for 45 s on ice. The protein was released from the cell debris using 8 M urea and dialyzed in 10 mM Tris-HCl (pH8.1) with 0.1% Triton X-100. The samples were stored at -20 °C prior to immunization.

**Generation of polyclonal antiserum and immunoprecipitation assays**. Rabbits were immunized with recombinant AcAS according to the manufacturer's standard procedures (Eurogentec, Seraing, Belgium). Reactivity of serum was compared to pre-immune serum using an enzyme-linked immunosorbent assay. Briefly, plates were coated with 100 ng antigen per well and a dilution range of serum (pre-immune versus serum from final bleed) was added. Antibody binding was

measured with a biotinylated goat anti-rabbit secondary antibody using the Vectastain ABC kit (Vector Labs). Immunoglobulins were absorbed on protein A/G sepharose (Pierce) and used to isolate AcAS from *P. falciparum* parasite lysates.

**Parasite lysates for AcAS assay**. Asynchronous blood-stage *P. falciparum* strain NF54, NF54-HGL or the T648M mutant line generated using CRISPR-Cas9 were released from RBCs by incubation with 0.06% saponin in PBS for 5 min on ice. Parasites were pelleted by centrifugation (10 min at 4,000xg), washed with PBS and lysed in 50 mM NaF, 20 mM Tris-HCl (pH 7.5), 0.1% Triton X-100, 2 mM dithiothreitol, 2 mM EDTA and 1% (v/v) Halt Protease Inhibitor Cocktail (Thermo-Fischer Scientific, Waltham, MA, USA). Suspensions were then sonicated 6 times for 3 s at an amplitude of 16 microns peak-to-peak. Sonicated samples were centrifuged at 24,000 g for 5 min at 4 °C and supernatants were used in immunoprecipitation and enzyme activity assays.

**AcAS assay**. Acetyl-CoA synthetase activity was measured using a radioactively labeled AcAS assay. The reaction mixtures contained 8 mM MgCl$_2$, 2 mM ATP, 30 μM Coenzyme A, 200 μM $^{14}$C-labeled sodium acetate (PerkinElmer), 50 mM Hepes-KOH, pH 8.5 and immunoprecipitated AcAS in a total volume of 35 μl. Reactions were incubated at 37 °C for 30 min. The reaction was terminated with 3.5 μl of a 10% acetic acid solution in 90% ethanol. Samples were loaded on DEAE filter paper (GE Healthcare) and washed thoroughly in 2% acetic acid solution in 95% ethanol to wash away unreacted acetate. After the discs were dried, they were transferred into scintillation vials containing 3 ml ScintiSafe 30% Cocktail (Fischer Scientific, Hampton, NH, USA). Radioactivity in each vial was counted using a Tri-Carb 2900TR Liquid Scintillation Analyzer (Packard Bioscience, Boston, MA). To test the inhibitory properties of MMV693183, 4'P-MMV693183, and CoA-MMV693183 on AcAS, a dilution range of the compound was pre-incubated for 30 min with the immunoprecipitated AcAS before initiation the AcAS reaction by adding the reaction mixture. The 4'P-MMV693183 and CoA-MMV693183 metabolites were obtained from Syncom, Groningen, the Netherlands.

**Cellular Thermal Shift Assay**. A cellular thermal shift assay (CETSA) was performed on infected RBCs or parasite lysates as described previously[31].

For CETSA on infected RBCs, synchronized trophozoite cultures (NF54-HGL parasites) were purified using magnetic-activated cell sorting (MACS). Parasites were resuspended in 1× PBS, aliquoted in PCR tubes (1.8×10$^7$ cells/tube) and subjected to 37 °C or 51 °C for 3 min on a pre-heated PCR machine, followed by 4 °C for 3 min. Parasites were mixed with 2× lysis buffer (100 mM HEPES, 10 mM β-glycerophosphate, 0.2 mM activated Na$_3$VO$_4$, 20 mM MgCl$_2$, with EDTA-free protease inhibitor cocktail (Merck)), and subjected to three freeze-thaw cycles using liquid nitrogen, followed by mechanical shearing using a syringe with a 25 G needle. Samples were spun down at 18,000 g for 20 min at 4 °C and the soluble fraction was flash-frozen in liquid nitrogen and stored at -80 °C.

For a CETSA on parasite lysates, synchronized trophozoite cultures (ACS11-tagged parasites) were treated with 0.1% saponin to lyse the RBCs, and washed three times in PBS. Subsequently, parasite pellets were resuspended in 1× lysis buffer and subjected to three flash-freeze-thaw cycles using liquid nitrogen, followed by mechanical shearing using a syringe with a 25 G and a 30 G needle. Samples were spun down at 18,000 g for 20 min at 4 °C. The supernatant was diluted to 2.1 mg/ml protein concentration, and 100 μl was added to each PCR tube containing 1 μl of the compound at a 100× concentration (final concentration of 1 μM). Samples were incubated for 30 min at room temperature, subjected to a thermal gradient for 3 min on a pre-heated PCR machine, followed by 4 °C for 3 min. Samples were spun down at 18,000 g for 20 min at 4 °C and the soluble fraction was flash-frozen in liquid nitrogen and stored at −80 °C.

**Western blot**. Samples were loaded on an 8% SDS-Page gel (Genscript) with 4×10$^6$ infected RBCs or 50 μg protein for the parasite lysate approach. Proteins were transferred to a nitrocellulose membrane that was blocked with 5% skim milk (Sigma) in PBS overnight and incubated with rabbit antiserum against AcAS (1:1000) for 1 h. Subsequently, membranes were washed three times with PBS-Tween for 5 min, followed by incubation with secondary horseradish peroxide (HRP)-conjugated goat anti-rabbit antibodies (Dako P0448, 1:1000) for 1 h. Blots were then washed three times with PBS-Tween for 5 min and twice with PBS. Following a 5-min incubation with Clarity Max Western ECL Substrate (BioRad), protein blots were imaged using the ImageQuant LAS4000 (GE Healthcare). The band intensity was quantified using Fiji software.

**Immunofluorescence microscopy**. Asynchronous asexual blood-stage AcAS-GFP or wild-type NF54 parasites were allowed to settle on poly-L-lysine coated coverslips for 20 min at room temperature. Parasites were fixed with 4% EM-grade paraformaldehyde and 0.0075% EM-grade glutaraldehyde in PBS for 20 min and permeabilized with 0.1% Triton X-100 for 10 min[78]. Samples were blocked with 3% bovine serum albumin (BSA) (Sigma-Aldrich) in PBS for 1 h. Samples of AcAS-GFP and NF54 parasites were incubated with primary chicken anti-GFP antibody (1:100, Invitrogen), or AcAS pre-immune and immune serum (1:500), respectively, in 3% BSA/PBS for 1 h, followed by incubation with secondary goat anti-chicken AlexaFluor 488 antibody (1:200, Invitrogen) in 3% BSA/PBS for 1 h. Nuclei were

visualized with 1 μM DAPI in PBS for 1 h. PBS washes were performed between different steps. Coverslips were mounted with Vectashield (Vector Laboratories). Images were taken with a Zeiss LSM880 Airyscan microscope with 63x oil objective with 405 and 488 nm excitations. Images were Airyscan processed before analysis with Fiji software. Since no quantitative comparisons were performed, brightness and contrast were slightly altered in Fiji to improve visualization of AlexaFluor488 and DAPI signals.

**Statistics and Reproducibility**. Dose-response assays were analyzed by a non-linear regression using a four-parameter model and the least squares method to find the best fit. One-way Analysis of variance (ANOVA) was performed using the Bonferroni's Multiple Comparison Test. For the majority of experiments, 2-4 biological replicates were performed. Typically, 2 replicates were performed for screening activities, and a minimum of 3 replicates are performed for other experiments unless stated otherwise.

**Reporting summary**. Further information on research design is available in the Nature Research Reporting Summary linked to this article.

## Data availability

The full metabolomics datasets are publicly available on the Metabolomics Workbench database under ST001985. The data associated with this study are presented in the paper, supplementary information and Source Data file. Genetically engineered parasite lines that were generated for this study are available upon request under a material of transfer agreement. Source data are provided with this paper.

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

## Acknowledgements

We gratefully acknowledge S. Mok for assistance with whole-genome sequencing analysis, C. Bioni for providing access to the laboratory for the Brazilian-field isolates ex vivo assessments, O. Byaruhanga, S. Orena, M. Okitwi and T. Katairo for assistance with ex vivo assays on fresh *P. falciparum* isolates in Uganda, S. Sax for technical assistance with the SCID mouse *P. falciparum* in vivo efficacy studies performed at Swiss TPH. We thank D.F. Wirth, A.K. Lukens and R. Summers for pre-publication sharing of data and fruitful discussions. T. Spielmann is acknowledged for providing the plasmid SLI-TGD. We also thank the Huck Institutes of Life Sciences Metabolomics Core Facility at Penn State University. LEdV was supported by a PhD fellowship from the Radboud Institute for Molecular Life Sciences, Radboudumc (RIMLS015-010), JMJV by an individual Radboudumc Master-PhD grant, TWAK by the Netherlands Organisation for Scientific Research (NWO-VIDI 864.13.009), JM by an NIH training grant (T32 DK120509), ML by the Bill & Melinda Gates Foundation (OPP1054480), JCN by the Bill & Melinda Gates Foundation (OPP1162467 and OPP1054480), JB by an Investigator Award from Wellcome (100993/Z/13/Z), DAF by the Medicines for Malaria Venture (RD/08/0015), the Department of Defense (W81XWH1910086) and the NIH (R01 AI109023), RVCG by Sao Paulo Research Foundation (FAPESP - CEPID grant 2013/07600-3 and 2020/12904-5), and ACCA by an Investigator Award from FAPESP (2019/19708-0). DGFA screening was supported by the Medicines for Malaria Venture (RD-08-2800, award to JB and AC), clinical field isolates experiments in Brazil were funded through ongoing MMV support, project RD-16-1066 (RVCG, ACCA), ex vivo studies in Uganda were supported by the NIH (R01AI139179) and Medicines for Malaria Venture (RD/15/0001), the efficacy study on ACT-resistant isolates was supported by the Medicines for Malaria Venture. We further acknowledge support by MalDA (OPP1054480; PI Dr. E. Winzeler, UCSD).

## Author contributions

T.W.A.K, and K.J.D. conceived of the work and did the overall supervision and analysis of parasitology, biochemistry, and molecular biology. L.E.d.V. performed and analyzed molecular biology experiments and in vitro parasitology assays and generated parasite mutants, T.W.A.K. provided supervision. J.M.J.V. generated parasite mutants and performed and analyzed immunofluorescence assays, L.E.d.V. and T.W.A.K provided supervision. P.A.M.J. performed and analyzed molecular biology and biochemistry assays, J.S. provided supervision. C.B. and A.F. generated and analyzed pharmacokinetic - pharmacodynamic models. J.M. performed and analyzed metabolomics data and whole genome sequencing data, M.L. provided supervision. S.W., M.B.J.-D., and I.A.-B. performed and analyzed efficacy and pharmacokinetics studies in SCID mice. C.F.A.P performed and analyzed growth assays on knockdown parasite mutants, J.C.N. provided supervision. L.B., J.M.B., R.W.M.H., T.H. K.M.J.K. performed and analyzed in vitro parasitology assays, K.J.D. provided supervision. K.R., J.S., T.Y. generated resistant parasite lines, and performed and analyzed in vitro parasitology experiments and whole-genome sequencing, D.A.F. provided supervision. G.T. designed and analyzed experiments to predict human PK parameters. B.C.F. performed and analyzed the parasite reduction rate assay, L.M.S., F.J.G. provided supervision. A.C. performed and analyzed the dual gamete formation assay, J.B. provided supervision. R.R. designed and analyzed the hemolytic toxicity assay. A.C.C.A., D.B.P., P.K.T. performed and analyzed ex vivo parasitology assays, R.V.C.G. R.A.C., P.J.R. provided supervision. C.R. and B.W. performed and analyzed parasitology assays using ACT-resistant field isolates. C.D.G. and N.A. advised on the toxicity studies. N.A. generated and analyzed CYP reaction phenotyping and CYP induction assays. P.H.H. overall supervised medicinal chemistry. R.B., B.C., R.W.S., and

J.S. advised on parasitology and drug development. L.E.d.V., T.W.A.K. and K.J.D. wrote the manuscript. All authors proofread and edited the manuscript.

## Competing interests

KJD and RWS hold stock in TropIQ Health Sciences B.V. PHHH is a consultant for TropIQ Health Sciences B.V, RB is a consultant for, and C.B., N.A., C.D.G., A.F., B.C., are employed by Medicines for Malaria Venture. Part of the data presented in this manuscript are included in patent application EP3674288A1, in the name of Medicines for Malaria Venture and with JS, PHHH, KJD and RVB as inventors. The remaining authors declare no competing interests.

## Additional information

[1]Department of Medical Microbiology, Radboudumc Center for Infectious Diseases, Radboud Institute for Molecular Life Sciences, Radboud University Medical Center, Nijmegen, The Netherlands. [2]Department of Dermatology, Radboud Institute for Molecular Life Sciences, Radboud University Medical Center, Nijmegen, The Netherlands. [3]Medicines for Malaria Venture, Geneva, Switzerland. [4]Department of Chemistry and Huck Center for Malaria Research, The Pennsylvania State University, University Park, PA, USA. [5]Department of Biological Engineering, Massachusetts Institute of Technology, Cambridge, MA, USA. [6]Department of Microbiology & Immunology, Columbia University Irving Medical Center, New York, NY, USA. [7]TropIQ Health Sciences, Nijmegen, The Netherlands. [8]Infectious Diseases Research Collaboration, Kampala, Uganda. [9]Sao Carlos Institute of Physics, University of São Paulo, São Carlos, São Paulo, Brazil, São Carlos, SP, Brazil. [10]The Art of Discovery, Derio, Spain. [11]Department of Life Sciences, Imperial College London, South Kensington, London, United Kingdom. [12]Global Health, GlaxoSmithKline, Tres Cantos, Madrid, Spain. [13]Research Center for Tropical Medicine of Rondonia, Porto Velho, Brazil. [14]Department of Immunology and Microbiology, University of Colorado Anschutz School of Medicine, Aurora, CO, USA. [15]Malaria Molecular Epidemiology Unit, Institut Pasteur du Cambodge, Phnom Penh, Cambodia. [16]Malaria Translational Research Unit, Institut Pasteur, Paris & Institut Pasteur du Cambodge, Phnom Penh, Cambodia. [17]Sygnature Discovery, Nottingham, United Kingdom. [18]Swiss Tropical and Public Health Institute, Basel, Switzerland. [19]University of Basel, Basel, Switzerland. [20]Department of Natural Sciences and Mathematics, Dominican University of California, San Rafael, CA, USA. [21]Department of Medicine, University of California, San Francisco, CA, USA. [22]Hermkens Pharma Consultancy, Oss, The Netherlands. [23]Center for Malaria Therapeutics and Antimicrobial Resistance, Division of Infectious Diseases, Department of Medicine, Columbia University Irving Medical Center, New York, NY, USA. [24]Department of Biochemistry & Molecular Biology, The Pennsylvania State University, University Park, PA, USA. [25]Present address: Department of Immunology and Infectious Diseases, Harvard T.H. Chan School of Public Health, Boston, MA, USA. [26]These authors contributed equally: Taco W. A. Kooij, Koen J. Dechering. ✉email: taco.kooij@radboudumc.nl; k.dechering@tropiq.nl

