## [Peer Review File · Nature Communications]

Peer review comments, initial review –

Reviewer #1 (Remarks to the Author):

This large amount of work from Dr Kooij and colleagues assembles the preclinical characterization of a small molecules from the family of pantothenamide i.e. MMV693183 for the cure and transmission blocking of malaria.

While the data are robust, fundamentally, the work on this series is not novel (target and series) and is rather an evolution of previous work that leads to a conclusive MoA and a potential drug candidate.

A few points would need further clarification such as the assessment of resistance in more details especially in vivo as well as the potential consequences in the field.

A better picture of the selectivity of such antiplasmodial agents towards other pathogens (protozoa, bacteria) would be useful to better identify the selectivity. Along these lines a clearer picture of the safety e.g. NOEL and margins that the candidate offer would be beneficial to the manuscript.

Finally, better statements on what drives efficacy in preclinical models would also be appreciated. Consequently, this report does not warrant publication in Nature as it relates to more specialized audience in the pharmaceutical field and lacks novelty.

Comments:

Abstract:

Line 56 “exceptional” is too subjective and description of the IC50 e.g. low nM activity / single digit nM activity would be more appropriate.

Line 59: “absorbed” rather than “adsorbed”?

Line112: please indicate incubation time for Plasmodium growth inhibition.

Line114: did authors only try single oral dose of 50 mg/kg. Any reasons for not doing a full dose response with monitoring of the recrudescence? Please clarify

Line119: did authors investigate at parasite recrudescence?

Line140: did authors investigate selection of mutants on field isolates?

Line196: please state the relevance of these cell lines/PHH/rat hepatocyte with respect to the mode of action.

Line199: In all assays i.e. mentioned above?

Line204: Since authors performed a 7-day tolerability study in rat, a NOEL should be mentioned. Why did authors not run a standard dose-range finding experiment?

Line251: As solubility and permeability are mentioned, it would also make sense to add the BCS class.

Line260: “However, parasites recrudesced within 15 days after treatment”. Have the recrudescence parasites been sequenced to ensure no mutants were present?

Line265: It is unclear here what drives efficacy i.e. Cmax, AUC, Cmin.

Also Table S15 is not referring to the experiment – please recheck numbering.

Line271: Authors indicate in their forward genetics experiments that the resistant parasites showed a 13-77 fold EC50 shift against MMV693183 in an asexual blood-stage growth assay. In their local sensitivity analysis performed on EC50, the modeling captures only a 10-fold shift. Arguably it would make sense to further expand this analysis to a relevant range i.e. the range found in the in vitro MIR experiment (13-77-fold shift).

Line378: please indicate a reference or disclose a generic synthetic route for PanAms.

Table S2: Please use either ng/mL or uM as units.

Authors contribution: "A. Fuchs for the PKPD analyses of pantothenamides that guided the selection of the preclinical candidate". Interestingly this seem to be more a co-authorship contribution level rather than an acknowledgement.

Conflict of interest statement: As MMV seem to have stakes in the patent – all related co-authors should be listed.

Reviewer #2 (Remarks to the Author):

This is a comprehensive work on a new preclinical lead that targets acetyl-CoA synthase through an antimetabolite mechanism. For over a decade, inhibitors against this pathway have been studied and the current work finally delivers a preclinical lead. The work is nicely written, the scientific rigor is high with carefully designed experiments including proper controls. Below are specific points that would require some clarification and potentially some experiments that will make some of the claims stronger and more solid.

Concerns, corrections and suggestions:

- 1) Figure 1a: It is more informative to show the full profile (curve) of the PK within 24 hs. As presented, it is not possible to observed the T_{max} and fitting of the curves supporting improved liver metabolism (at this point of the reading).
- 2) Line 109-111: the reference provided is a patent, therefore, a summary of the SAR in this paper would be appropriated. Are compounds reported in table 1 the only ones synthetized or why only those are reported?
- 3) Line 115: please indicate the half-life (t_{1/2}) for each compound in NSG mice to support the statement "blood concentrations decreased rapidly over time".
- 4) Line 119-121: this statement would be better supported by assessing the Clint in mouse hepatocytes and compare values with their respective t_{1/2}. In addition, one of the major speculations is that clearance by the liver would play an important role in the human PK for this compound with a striking change in the predicted t_{1/2} (~30-fold increase). Thus, pair-wise comparisons within same species may better support those statements. The hypothesis that reduced liver metabolism in humans would translate into longer t_{1/2} could be in part assessed using NSG mice engrafted with human liver cells as it has been shown to be a good model to predict human PK in cases where liver metabolism plays an important role.
- 5) The parasite reduction rate assay indicates that parasites will need to be continuously exposed to drug for 48 hs in order to clear infection. This is not very impressive (unfortunately) and explains why failed to clear parasitemia in vivo (recrudescence after 15 days) using NSG mice where MMV693183 was below 40 nM within 24 h. The idea that MMV693183 can still be proposed as a single oral dose is too speculative with little experimental support.
- 6) Among the CoA-dependent metabolism is the energy supply/ETC and a recent publication from Dr. Fidock's group (co-author here) shows that presence of K13 mutations alters energy supply/ETC. What is the EC₅₀ of MMV693183 in artemisinin-resistance parasites?
- 7) Is MMV693183 stage specific in asexual blood stages? At which stage of the asexual intraerythrocytic cycle parasites die? A representative Giemsa-stained smear of treated cultures would be informative following treatment over time.
- 8) How fast resistance was generated? Briefly described how drug exposure was performed and for how long.
- 9) Line 152: "conferred cross-resistance to other PanAms." But only two compounds are shown. Is

there a structure-resistance relationship that could have resulted in a different preclinical candidate or to improve the lead to avoid resistance?

10) Please report the chemical structure of CXP18.6-052 and its EC50.

11) Line 154-155: please, a representative MS profile showing how the structure was identified and confirmed is strongly encourage.

12) Line 184-187: why immunofluorescence microscopy was used instead of live parasites with GFP? Colocalization with other sub-cellular markers would strengthen the analysis and statement such as an ER marker (BiP) and Golgi.

13) Table S17: concentration in urine for animal 2 is five-times higher than the other two animals while other values were quite similar. Is this correct?

14) Figure S14: why values below the LLOQ are reported? How those values were accurately calculated? Also, please clarify what 3x2.5 mg/kg, 6x2.5 mg/kg and 6x5 mg/kg means. It is not explained in the legend/method section.

15) Methods for sample preparation and LC/MS/MS analysis for the mice study (lines 468- 471) need to be fully described unless published before where please indicate a reference.

Minor corrections/suggestions:

1) Table 1: report the SEM for all calculated IC50 values and indicate what N/A means.

2) Figures in the supplemental material may benefit of higher resolution (currently pixilated)

3) Figure S3: symbols are difficult to identify, using different contrasting colors would be ideal.

4) Legend Fig 5: LLOQ is described but not shown in the figures.

5) Line 395: clarify if assays were performed with synchronous or asynchronous cultures.

6) Figure S3: indicate the concentrations of artemisinin, chloroquine and atovaquone.

7) Drug vehicle used in dogs is not reported, why?

Reviewer #3 (Remarks to the Author):

This manuscript prevents the discovery of and characterization of a potential new antimalarial clinical candidate that targets the CoA pathway, which has been of interest in malaria drug discovery for well over 60 years without delivering a drug candidate. The final compound discussed herein is the first such inhibitor viable for further development. Overall, the manuscript presents a technically well executed set of experiments that generally support the assertions made and that yield a reasonably well validated clinical candidate which is a significant finding. However, the novelty is undercut by the authors prior disclosures in the area and the clinical candidate looks to fall performance wise in the middle of the current portfolio.

The manuscript describes six new "PanAm" compounds that are derivative of those they have previously reported but sufficiently novel to warrant publication. Neither the manuscript nor the supplemental materials contain sufficient descriptions of the synthesis or characterization of the novel molecules. To meet accepted standards the authors need to provide the synthetic procedures at a sufficient level of detail to allow others to synthesize the molecules and stringent data concerning both identity and purity of the molecules used in the biological experiments (minimally to include ¹H and ¹³C NMR, HRMS, HPLC traces; ideally to include elemental analysis or melting point or DSC, IR, UV, etc).

The basic performance of the final compound discussed ticks all the boxes for a quality clinical

candidate including: good potency in vitro (10 nM asexual / 30 nM sexual) against a wide range of strains/isolates of Plasmodium; reasonable potency in vivo in the current gold standard SCID model (50 mg/kg dose providing clearance within 3 days, and 2 weeks of suppression); a rapid in vitro PRR; good tolerability and selectivity for the pathway in host vs pathogen; reasonable pharmacokinetics; and reasonable predicted human PK (35 h half life) and dose (15-650 mg). However, there are some significant holes in the presentation. First, the experiments to give an estimated potency in the SCID model were carried out but no ED90, MIC, or MPC is presented – the authors should present these parameters and discuss in comparison to the rest of the portfolio. Second, the in vivo PRR is not presented or discussed; from the data that are presented it seems that the in vivo PRR is more in the medium than the fast category (more pyrimethamine like). The data for this critical parameter need to be presented and, if it is truly significantly different from the in vitro results, the difference needs to be discussed. Finally, the overall performance looks more congruent with a 3-day drug than a 1-day drug and in particular the dose prediction is highly variable and sensitive to clearance (ranging from quite good (15 mg) to a non-starter (650 mg)). The authors should give a more balanced discussion of this assessment. If any data exist to mitigate the risk around clearance (for example hepatocyte clearance studies) that should be presented.

The mechanism studies provide a solid underpinning for the concept that the compounds target ACS within the pathway. The drug induction of a resistance conferring mutation that maps to ACS and is confirmed both by knock-in to a native locus giving cross resistance and conditional knockout increasing sensitivity very unambiguously shows that ACS is a resistance locus for the drug. The CETSA data clearly show the potential for a metabolite of the drug to bind to ACS and provide a plausible target interaction hypothesis; however the 15-fold weaker inhibitory potency on immunoprecipitated protein, relative to the cellular activity, needs to be explained. These data are suggestive but leave open the possibility that the actual inhibitor is a different metabolite and do not disprove the hypothesis that the parent drug binds to a different target. To fully resolve this the authors need some alternative approaches: feeding / rescue experiments to show the putative metabolite is active; heterologous expression and binding/inhibition studies of ACS; or other direct evidence of target. In absence of this, the discussion should be more fully caveated.

REVIEWER COMMENTS

Reviewer #1 (Remarks to the Author):

This large amount of work from Dr Kooij and colleagues assembles the preclinical characterization of a small molecules from the family of pantothenamide i.e. MMV693183 for the cure and transmission blocking of malaria.

While the data are robust, fundamentally, the work on this series is not novel (target and series) and is rather an evolution of previous work that leads to a conclusive MoA and a potential drug candidate.

We thank the reviewer for acknowledging the amount of work and robustness of the study. Naturally, this work builds upon previous work, as is generally the case in research, providing a critical next step in the development of a new anti-malarial.

A few points would need further clarification such as the assessment of resistance in more details especially *in vivo* as well as the potential consequences in the field.

A better picture of the selectivity of such antiplasmodial agents towards other pathogens (protozoa, bacteria) would be useful to better identify the selectivity.

Other work has focused on studying pantothenamides against bacteria. We have added references to these studies in the introduction. Furthermore, one of our previously tested pantothenamide (CXP18.6-017) has shown micromolar activity against *T. gondii*, which is also added to the introduction in this revised version.

Along these lines a clearer picture of the safety e.g. NOAEL and margins that the candidate offer would be beneficial to the manuscript.

In the revised version of the manuscript we have included the results of extensive safety studies in rats, including PK, maximum tolerated dose and dose range finding studies, and have indicated provisional safety margins based on the rat data.

Finally, better statements on what drives efficacy in preclinical models would also be appreciated.

In the *in vivo* efficacy studies, we observed a potential mixed behavior concerning what drives MMV693183 efficacy. On the one hand, the efficacy of single dose regimens seem to endure beyond the time where the concentrations went below the LLOQ, possibly related to a C_{max} effect. And on the other hand, recrudescence was delayed in mice treated with multiple dose regimens, possibly related to an AUC or time above certain target concentration effect.

Taking advantage of the rich PKPD data generated, we developed a PKPD model to fully characterize the relationship between MMV693183 concentration and killing rate, using an E_{max} model. This PKPD model considers the effect of the overall concentration profile on the killing effect of MMV693183. It also allowed us to simulate the expected PK and PD profile at any MMV693183 dose regimen and consequently predict the efficacious dose to achieve an 9 or 12 log reduction in parasitemia.

In summary, efficacy is driven by the concentration-dependent killing rate of the compound. We have included a more elaborate description of the PD model in the revised version of the manuscript.

Consequently, this report does not warrant publication in Nature as it relates to more specialized audience in the pharmaceutical field and lacks novelty.

Comments:

Abstract:

Line 56 "exceptional" is too subjective and description of the IC50 e.g. low nM activity / single digit nM activity would be more appropriate.

This is changed in the abstract

Line 59: “absorbed” rather than “adsorbed”?

This is adjusted in the abstract

Line112: please indicate incubation time for Plasmodium growth inhibition.

The incubation time is added to the legend of Table 1

Line114: did authors only try single oral dose of 50 mg/kg. Any reasons for not doing a full dose response with monitoring of the recrudescence? Please clarify

The single dose experiment was intended as an initial survey to investigate to what extent the new compounds show a prolonged killing effect *in vivo* (that outlives blood exposure) as observed for pantothenamide MMV689258 in our previous study (PMID 31534021). Therefore, one single dose was chosen to test the efficacy and pharmacokinetics before the most promising compound was progressed to additional studies. Investigating recrudescence was part of these later studies. This strategy is clarified in the new version of the manuscript.

Line119: did authors investigate at parasite recrudescence?

As explained above, the initial *in vivo* experiment was intended to survey PK-PD and select a late lead for follow up studies.

Line140: did authors investigate selection of mutants on field isolates?

Selection of mutants was performed using laboratory strains NF54 and Dd2 that allowed establishment of a minimum inoculum of resistance using standardized protocols. The lines we used have very distinct geographical origins and, in our view, adding additional *in vitro* selection experiments would add no additional insights to predict emergence of resistance *in vivo* in field settings. Possible pre-existing resistance in the field has been investigated by interrogation of the pf3k database from the malariaGEN study. The T648M mutation in ACS has not been identified in 2512 samples received from 14 different countries. This suggests that there is no pre-existing resistance. We have referred to this study in our discussion.

Line196: please state the relevance of these cell lines/PHH/rat hepatocyte with respect to the mode of action.

A previous study by Zhang et al. 2007 showed that targeting the CoA pathway results in liver damage. Therefore, we specifically studied the effect of pantothenamides on liver cells. A HepG2 cell line is a standard cell line to test the cytotoxicity of compounds. However, since cell lines may be less metabolically active, we also studied the cytotoxicity of pantothenamides in primary hepatocytes cells. We have added this reasoning to the main text in paragraph ‘MMV693183 safety’.

Line199: In all assays i.e. mentioned above?

‘All assays’ refers to the sentence before. We have changed this to ‘In this cross-reactivity assay, inhibition was <50% at a test concentration of 10 μ M (Table S24).’

Line204: Since authors performed a 7-day tolerability study in rat, a NOEL should be mentioned. Why did authors not run a standard dose-range finding experiment?

Our initial submission included a preliminary experiment to test the effect of MMV693183 on blood glucose, as the Zhang, 2004 study indicated a severe drop in glucose following inhibition of the pantothenate-CoA pathway. At the request of the reviewer, we have now included three more studies in rats, including a DRF study. The DRF study indicated a NOEL of 30 mg/kg/day for eight days and a safety window >30 fold compared to the predicted human efficacious exposure.

Line251: As solubility and permeability are mentioned, it would also make sense to add the BCS class. MMV693183 belongs to BCS class I, we have added this to the main text.

Line260: “However, parasites recrudesced within 15 days after treatment”. Have the recrudescence parasites been sequenced to ensure no mutants were present?

The recrudescence is perfectly captured by our PK-PD model and the result of the fast clearance of the compound in mice as described in the ‘Pharmacokinetic properties’ section, which is expected to be significantly slower in humans as described in the section ‘Human pharmacokinetic predictions’. Furthermore, it is expected to find mutations *in vivo*, as *in vitro* resistance methods predict the development of resistance in a mouse model (Mandt *et al.*, PMID: 31801884). The study of emergence of resistance *in vivo* was outside of the scope of the current study but the subject of our

ongoing and future studies, including anticipated studies in human volunteers. We have indicated this in the revised manuscript.

Line265: It is unclear here what drives efficacy i.e. C_{max}, AUC, C_{min}.

As explained above, our PK-PD modeling indicates that efficacy is driven by the killing rate of the compound as a function of the plasma concentration in time.

Also Table S15 is not referring to the experiment – please recheck numbering.

We do refer to Table S15 (Table S12 in the revised manuscript) in the main text and the numbering is correct according to us. We have added “*in vitro* human hepatocyte relay assay” in the name of the graph to refer to the text.

Line271: Authors indicate in their forward genetics experiments that the resistant parasites showed a 13-77 fold EC₅₀ shift against MMV693183 in an asexual blood-stage growth assay. In their local sensitivity analysis performed on EC₅₀, the modeling captures only a 10-fold shift. Arguably it would make sense to further expand this analysis to a relevant range i.e. the range found in the *in vitro* MIR experiment (13-77-fold shift).

We have extended the sensitivity analysis up until a 100-fold change and included this in the section ‘Prediction of the human efficacious dose using a PKPD model’.

Line378: please indicate a reference or disclose a generic synthetic route for PanAms.

We have added a description of the synthesis of all PanAms in the Supplementary Methods

Table S2: Please use either ng/mL or μ M as units.

We have now listed the data in μ M.

Authors contribution: “A. Fuchs for the PKPD analyses of pantothenamides that guided the selection of the preclinical candidate”. Interestingly this seems to be more a co-authorship contribution level rather than an acknowledgement.

This is a fair point and we have included A. Fuchs as a co-author

Conflict of interest statement: As MMV seem to have stakes in the patent – all related co-authors should be listed.

This has been clarified in the conflict of interest statement.

Reviewer #2 (Remarks to the Author):

This is a comprehensive work on a new preclinical lead that targets acetyl-CoA synthase through an antimetabolite mechanism. For over a decade, inhibitors against this pathway have been studied and the current work finally delivers a preclinical lead. The work is nicely written, the scientific rigor is high with carefully designed experiments including proper controls. Below are specific points that would require some clarification and potentially some experiments that will make some of the claims stronger and more solid.

We thank the reviewer for the kind words and for acknowledging the quality of our work.

Concerns, corrections and suggestions:

1) Figure 1a: It is more informative to show the full profile (curve) of the PK within 24 hs. As presented, it is not possible to observe the T_{max} and fitting of the curves supporting improved liver metabolism (at this point of the reading).

We have adjusted the figure to show the PK values within 24 hours.

2) Line 109-111: the reference provided is a patent, therefore, a summary of the SAR in this paper would be appropriate. Are compounds reported in table 1 the only ones synthesized or why only those are reported?

A SAR analysis section has been added to this paragraph and examples of modifications have been added in supplementary Table S1 and S2. The six compounds presented in this manuscript were

chosen based on their potency and stability in hepatocytes and this is clarified in the revised manuscript.

3) Line 115: please indicate the half-life ($t_{1/2}$) for each compound in NSG mice to support the statement “blood concentrations decreased rapidly over time”.

We have added the $t_{1/2}$ lives of each compound in table S1.

4) Line 119-121: this statement would be better supported by assessing the Clint in mouse hepatocytes and compare values with their respective $t_{1/2}$. In addition, one of the major speculations is that clearance by the liver would play an important role in the human PK for this compound with a striking change in the predicted $t_{1/2}$ (~30-fold increase). Thus, pair-wise comparisons within same species may better support those statements. The hypothesis that reduced liver metabolism in humans would translate into longer $t_{1/2}$ could be in part assessed using NSG mice engrafted with human liver cells as it has been shown to be a good model to predict human PK in cases where liver metabolism plays an important role.

Indeed, we assume that the liver plays an important role in elimination of PanAms in humans. This is based on the observation that renal clearance plays a minor role in rats and dogs and the excellent pair-wise correlation between *in vitro* hepatocyte data and the non-renal clearance observed *in vivo* in rats and dogs. These data are presented in tables S15-17 and the ‘pharmacokinetic properties’ paragraph in the result section.

The use of NSG mice engrafted with human hepatocytes for PK studies would face the challenge that engraftment is never 100%, and that hepatic metabolism would be a mixture of mouse and human metabolism. Instead, we have predicted human hepatic metabolism based on allometry and, in a second scenario and given the excellent *vitro-vivo* correlation for rodents and dogs, on *vitro* metabolism studies in human hepatocytes. These analyses are presented in the ‘Human pharmacokinetic predictions’ paragraph in the result section.

5) The parasite reduction rate assay indicates that parasites will need to be continuously exposed to drug for 48 hs in order to clear infection. This is not very impressive (unfortunately) and explains why failed to clear parasitemia *in vivo* (recrudescence after 15 days) using NSG mice where MMV693183 was below 40 nM within 24 h. The idea that MMV693183 can still be proposed as a single oral dose is too speculative with little experimental support.

The parasite reduction assay presented in Figure S3 indicates that the number of viable parasites drops below the detection limit before the 48 hr timepoint. This indicates that MMV693183 is a very fast-acting compound and at par or even better than artemisinin, which is the fastest acting antimalarial described to date. Therefore, we do not understand the reviewer’s comment that the killing rate is not very impressive.

The recrudescence in mice can be explained by the observation that mice show fast clearance of the compound as described in the ‘pharmacokinetic properties’ paragraph and the ‘Prediction of the human efficacious dose using a PKPD model’ paragraph. In spite of the observation that parasite clearance continues whilst blood levels of the parent compound have dropped below the detection limit, exposure in mice is not sufficient to completely eliminate all parasites. Our data indicate that exposure in humans may be very different. Based on extensive cross-species PK studies, allometry and *in vitro-vivo* correlations, we estimate a human half-life of 32.4 hours in comparison to the half-life of 2.5 hours observed in mice (table S3). Based on the predicted PK and the PK-PD correlation derived from the humanized mice studies, we estimate that a single human dose of ≤ 30 mg would provide sufficient exposure to achieve a 12 log reduction in parasitemia. We feel that our approach is in line with the state-of-the-art and well substantiated. In addition, we discuss the limitations of the PK-PD model in the discussion section and have indicated that human studies are ultimately needed to verify the dose predictions described in our manuscript.

6) Among the CoA-dependent metabolism is the energy supply/ETC and a recent publication from Dr. Fidock's group (co-author here) shows that presence of K13 mutations alters energy supply/ETC. What is the EC₅₀ of MMV693183 in artemisinin-resistance parasites?

Following the suggestion of the reviewer, we have tested MMV693183 against *P. falciparum* isolated from patients that were adapted to *in vitro* culture. These were Cambodian isolates and are associated with a clinically relevant resistance to ACT drugs, including C580Y and Y493H mutation in Kelch13, pfmdr1 amplification, pfcr1 mutations and/or pfpm2 amplification. These parasites showed an IC₅₀ of 0.21-1.84 nM, which is similar to our IC₅₀ values observed for NF54. This data is presented in Figure S6 in the revised manuscript.

7) Is MMV693183 stage specific in asexual blood stages? At which stage of the asexual intraerythrocytic cycle parasites die? A representative Giemsa-stained smear of treated cultures would be informative following treatment over time.

We have tested whether MMV693183 has stage-specific activity and we found that it is active against both early and late stage asexual blood stages. We have included this data in Figure S4 in the Supplementary Information.

8) How fast resistance was generated? Briefly described how drug exposure was performed and for how long.

We have added this information in the results section in paragraph 'A role of AcAS in the mechanism of action of MMV693183' and more elaborate in the Methods section in paragraph 'Induction of drug-resistant parasites'..

9) Line 152: "conferred cross-resistance to other PanAms." But only two compounds are shown. Is there a structure-resistance relationship that could have resulted in a different preclinical candidate or to improve the lead to avoid resistance?

Following submission of our manuscript, a study was published showing that the T648M mutation selected by PanAms is also selected by chemically distinct AcAS inhibitors, indicating a common mechanism of resistance through a mutation in the active site (Summers et al. 2021). We have not extensively studied SAR against T648 mutants. Although it is possible that it would lead to identification of compounds with activity against the T648M mutant, it would not preclude the possibility that such a compound would select a different mutation leading to resistance. In the revised manuscript we have included a discussion of this mutant in the context of resistance.

10) Please report the chemical structure of CXP18.6-052 and its EC₅₀.

We have included a reference to our previous publication where we describe the synthesis and activity of CXP18.6-052.

11) Line 154-155: please, a representative MS profile showing how the structure was identified and confirmed is strongly encourage.

We have added a MS profile of the different metabolites of MMV693183 in Supplementary Figure S10.

12) Line 184-187: why immunofluorescence microscopy was used instead of live parasites with GFP? Colocalization with other sub-cellular markers would strengthen the analysis and statement such as an ER marker (BiP) and Golgi.

The microscopy facility in the institute where these experiments were performed do not have microscopes available within the safety level required for *P. falciparum*. Therefore, we fixed and stained the parasites to localize ACS. The staining pattern appears rather undefined intra-parasitic with no obvious subcellular localization, which is why we concluded taking the function into consideration that the protein is likely cytoplasmic and possibly ER. However, as we do not have access to the ER-marker BiP, or other relevant markers, to confirm/exclude such specific localizations, we have changed our interpretation "widespread, undefined intraparasitic". Most

importantly, our data do not seem to support earlier observations of a nuclear localization from Summer *et al.*, but rather the cytosolic localization that was identified in Prata *et al.* We have included references to both studies in the revised manuscript

13) Table S17: concentration in urine for animal 2 is five-times higher than the other two animals while other values were quite similar. Is this correct?

We have double checked the data, and this is indeed correctly presented.

14) Figure S14: why values below the LLOQ are reported? How those values were accurately calculated? Also, please clarify what 3x2.5 mg/kg, 6x2.5 mg/kg and 6x5 mg/kg means. It is not explained in the legend/method section.

Thanks for spotting this mistake. The LLOQ value was incorrectly set up in the figure. We have updated the figure and the legend to clarify that the values below the LLOQ were represented at the LLOQ value to allow graphical representation. The description of the dose regimens are also included in the updated legend.

15) Methods for sample preparation and LC/MS/MS analysis for the mice study (lines 468- 471) need to be fully described unless published before where please indicate a reference.

A full description of QC and sample preparation, HPLC-MS/MS conditions, data acquisition and processing is now provided in the Materials and Methods section.

Minor corrections/suggestions:

1) Table 1: report the SEM for all calculated IC₅₀ values and indicate what N/A means.

We have tested the activity of most compounds in two independent experiments, and are therefore not able to determine the SEM. To better indicate the range of IC₅₀s, we are now presenting the individual values of each replicate.

2) Figures in the supplemental material may benefit of higher resolution (currently pixilated)

Thank you, we have changed the figures that needed a higher resolution

3) Figure S3: symbols are difficult to identify, using different contrasting colors would be ideal.

We have changed the symbols and colors so it is easier to differentiate between the compounds.

4) Legend Fig 5: LLOQ is described but not shown in the figures.

Thank you for spotting this, we have removed the mention of LLOQ from the text since this is a simulation and not measured data.

5) Line 395: clarify if assays were performed with synchronous or asynchronous cultures.

We have added that these assays were performed on asynchronous parasites

6) Figure S3: indicate the concentrations of artemisinin, chloroquine and atovaquone.

We have added the concentrations in the legend

7) Drug vehicle used in dogs is not reported, why?

Thank you for spotting this mistake, we have added this in the revised version.

Reviewer #3 (Remarks to the Author):

This manuscript prevents the discovery of and characterization of a potential new antimalarial clinical candidate that targets the CoA pathway, which has been of interest in malaria drug discovery for well over 60 years without delivering a drug candidate. The final compound discussed herein is the first such inhibitor viable for further development. Overall, the manuscript presents a technically well executed set of experiments that generally support the assertions made and that yield a reasonably well validated clinical candidate which is a significant finding.

We thank the reviewer acknowledging the quality of our work and the significance of the new preclinical candidate.

However, the novelty is undercut by the authors prior disclosures in the area and the clinical candidate looks to fall performance wise in the middle of the current portfolio. The manuscript describes six new “PanAm” compounds that are derivative of those they have previously reported but sufficiently novel to warrant publication. Neither the manuscript nor the supplemental materials contain sufficient descriptions of the synthesis or characterization of the novel molecules. To meet accepted standards the authors need to provide the synthetic procedures at a sufficient level of detail to allow others to synthesize the molecules and stringent data concerning both identity and purity of the molecules used in the biological experiments (minimally to include ¹H and ¹³C NMR, HRMS, HPLC traces; ideally to include elemental analysis or melting point or DSC, IR, UV, etc).
We have included the synthetic procedures and show the identity and purity of these molecules in the Supplementary Information.

The basic performance of the final compound discussed ticks all the boxes for a quality clinical candidate including: good potency in vitro (10 nM asexual / 30 nM sexual) against a wide range of strains/isolates of Plasmodium; reasonable potency in vivo in the current gold standard SCID model (50 mg/kg dose providing clearance within 3 days, and 2 weeks of suppression); a rapid in vitro PRR; good tolerability and selectivity for the pathway in host vs pathogen; reasonable pharmacokinetics; and reasonable predicted human PK (35 h half life) and dose (15-650 mg). However, there are some significant holes in the presentation.

First, the experiments to give an estimated potency in the SCID model were carried out but no ED90, MIC, or MPC is presented – the authors should present these parameters and discuss in comparison to the rest of the portfolio.

Estimates of ED90, MIC and MPD are now presented in table S22. In addition, we have included a more elaborate discussion of the PK-PD model and its parameters in the revised version of the manuscript.

Second, the in vivo PRR is not presented or discussed; from the data that are presented it seems that the in vivo PRR is more in the medium than the fast category (more pyrimethamine like). The data for this critical parameter need to be presented and, if it is truly significantly different from the in vitro results, the difference needs to be discussed.

The *in vitro* PRR is based on viable parasites reduction, while the *in vivo* data only present the number of parasites quantified by flow cytometry, which does not discriminate between live and dead parasites. Therefore, we could not derive an *in vivo* killing rate. However, the PK-PD model presented in the manuscript does estimate *in vivo* parasite clearance and this is presented in table S21.

Finally, the overall performance looks more congruent with a 3-day drug than a 1-day drug and in particular the dose prediction is highly variable and sensitive to clearance (ranging from quite good (15 mg) to a non-starter (650 mg)). The authors should give a more balanced discussion of this assessment.

Indeed the dose prediction is sensitive to variation in clearance. We present two approaches to estimate human hepatic clearance: one based on allometry and one based on vitro clearance data. Both approaches predict a total dose <30 mg. In the original paper we presented a hypothetical scenario where clearance would be 5 times higher, leading to a dose of 650 mg. This scenario was purely hypothetical and merely to illustrate the sensitivity analyses that we present. We have now modified this discussion to present a more balanced view, including a discussion of the limitations of the PK-PD model, in the discussion section.

If any data exist to mitigate the risk around clearance (for example hepatocyte clearance studies) that should be presented.

We have measured the clearance of MMV693183 in different batches of hepatocytes and we presented the worst-case scenario in table 1. We have now included the data of all our clearance

assays (Table S18) and indicate that we have used the highest value to present a worst case scenario for the prediction of human hepatocytic clearance.

The mechanism studies provide a solid underpinning for the concept that the compounds target ACS within the pathway. The drug induction of a resistance conferring mutation that maps to ACS and is confirmed both by knock-in to a native locus giving cross resistance and conditional knockout increasing sensitivity very unambiguously shows that ACS is a resistance locus for the drug. The CETSA data clearly show the potential for a metabolite of the drug to bind to ACS and provide a plausible target interaction hypothesis; however the 15-fold weaker inhibitory potency on immune-isolated protein, relative to the cellular activity, needs to be explained.

We have added a few hypotheses in the discussion that may explain this difference in activity, including a potential activity of dP-CoA-PanAm, degradation of CoA-PanAm, difference in concentration of CoA and acetate in the assay compared to intracellular concentration, and the possible accumulation of the CoA-analog upon uptake as this metabolite cannot diffuse freely over the membrane.

These data are suggestive but leave open the possibility that the actual inhibitor is a different metabolite and do not disprove the hypothesis that the parent drug binds to a different target. To fully resolve this the authors need some alternative approaches: feeding / rescue experiments to show the putative metabolite is active; heterologous expression and binding/inhibition studies of ACS; or other direct evidence of target. In absence of this, the discussion should be more fully caveated.

We cannot say for sure if there is another active metabolite, such as the dP-CoA-PanAm, or another target. In the discussion we have added hypotheses that explain the difference between the AcAS activity and anti-parasitic activity of the compound, such as a possible additional activity of dP-CoA-PanAm. To further confirm that AcAS is targeted by CoA-PanAm, we have repeated the AcAS assay and included the T648M mutant. We now show that the T648M mutant is resistant to inhibition by CoA-PanAm compared to the wild-type AcAS (Figure 3b), confirming that AcAS is the target of CoA-PanAm. Unfortunately, CoA-PanAm is very unlikely to cross membranes, and therefore we cannot show the antiparasitic effect of this metabolite alone. Based on recent literature (Summers *et al.*), we know that AcAS is the drug target of chemically different compounds, and resistance to these inhibitors is mediated through the same T648M mutation in AcAS. This shows that targeting AcAS with CoA-PanAm is very likely to explain the lethal effect of PanAms.

Peer review comments, second review –

Reviewer #1 (Remarks to the Author):

This large amount of work from Dr Kooij and colleagues assembles the preclinical characterization of a small molecules from the family of pantothenamide i.e. MMV693183 for the cure and transmission blocking of malaria.

The manuscript has improved yet a few remaining points would need clarity.

Specifically, given the emphasis placed on developing new antimalarial in light of drug resistance and beyond the expected fact that no pre-existing resistance is to be observed in field isolates due to the new mode of action, it would nevertheless seem fair to disclose/discuss two points:

-Please disclose and comment the fitness costs of mutant parasites (e.g. cT648M) selected in vitro (Table S5) as this is a key parameter as to the likelihood to spread resistance in the field.

-There seem to be a disconnect between the in vitro selection of mutant i.e. no mutants selected at 1×10^8 and the *P. falciparum* SCID mouse model for which the parasite load is about the same and were recrudescence of parasites is seen – please comment.

Minor: line 104 “PanAms are metabolized” by? Please specify.

Line: 275: “At later time points, parasites recrudesced in all treatment groups (Fig S18).” Please specify mutant or WT.

Reviewer #2 (Remarks to the Author):

All previous concerns were addressed.

Reviewer #3 (Remarks to the Author):

The authors have responded well to all questions and criticisms raised during the first review. In particular the presentation of data underlying and discussion of human dose prediction is much stronger; likewise the additional data and discussion around target engagement really strengthen that portion. Overall this is an exciting and excellently put together manuscript.

REVIEWER COMMENTS

Reviewer #1 (Remarks to the Author):

This large amount of work from Dr Kooij and colleagues assembles the preclinical characterization of a small molecules from the family of pantothenamide i.e. MMV693183 for the cure and transmission blocking of malaria.

The manuscript has improved yet a few remaining points would need clarity.

Specifically, given the emphasis placed on developing new antimalarial in light of drug resistance and beyond the expected fact that no pre-existing resistance is to be observed in field isolates due to the new mode of action, it would nevertheless seem fair to disclose/discuss two points:

-Please disclose and comment the fitness costs of mutant parasites (e.g. cT648M) selected in vitro (Table S5) as this is a key parameter as to the likelihood to spread resistance in the field.

Our previously studied PanAm-resistant parasite with both a mutation in AcAS and ACS11 showed a fitness cost (Schalkwijk *et al.*, 2019), but we have not tested whether the T648M mutant has a similar growth defect. We disclosed this in the discussion. Furthermore, it is still challenging to predict the spread of resistance based on fitness cost, as the parasites may also adapt by generating compensatory mutations. Therefore, the major determinant to select new compound candidates is currently the minimum inoculum for resistance (MIR). An MIR of $\geq 10^9$ seems to be a sufficient high barrier to resistance while compounds with an MIR ≥ 7 may still be valuable candidates for combination therapies when resistance risk is managed (Duffey *et al.* 2021). For MMV693183 we determined the MIR at 10^9 indicating a sufficiently barrier for resistance development. We have added the following lines to the discussion to clarify (lines 358-361):

“The spread of resistance may also be affected by a possible fitness cost for resistant parasites, as was observed for the previously generated PanAm-resistant parasites with a mutation in AcAS and ACS11 (15). However, it is still unknown whether MMV693183-resistant parasites have a fitness defect. Fortunately, [...] studies. Reassuringly, the minimum inoculum for resistance development against MMV693183 was 9, which is considered to sufficiently reduce the risk of resistance against new antimalarials (54).”

-There seem to be a disconnect between the in vitro selection of mutant i.e. no mutants selected at 1×10^8 and the *P. falciparum* SCID mouse model for which the parasite load is about the same and were recrudescence of parasites is seen – please comment.

This could be explained by the faster clearance of the compound in mice. We refer to this in the discussion (lines 362-364). “The recrudescence observed in our *in vivo* efficacy experiments is in line with the fast clearance of the compound in mice and well captured by the PKPD model presented.”

Minor: line 104 “PanAms are metabolized” by? Please specify.

We changed the sentence to: ‘PanAms are metabolized by three enzymes of the CoA pathway’.

Line: 275: “At later time points, parasites recrudescenced in all treatment groups (Fig S18).” Please specify mutant or WT.

We clarified by changing the section to: ‘..., while 3D7 parasites were being cleared at four to six days. At later time points, these parasites recrudescenced in all treatment groups (Fig S18).’

Reviewer #2 (Remarks to the Author):

All previous concerns were addressed.

Reviewer #3 (Remarks to the Author):

The authors have responded well to all questions and criticisms raised during the first review. In particular the presentation of data underlying and discussion of human dose prediction is much stronger; likewise the additional data and discussion around target engagement really strengthen that portion. Overall this is an exciting and excellently put together manuscript.